**REPORT**

JCB Journal of Cell Biology

# Faa1 membrane binding drives positive feedback in autophagosome biogenesis via fatty acid activation

Verena Baumann[1,2]*, Sonja Achleitner[1,2,3]*, Susanna Tulli[1,2]*, Martina Schuschnig[1,2], Lara Klune[1,2], and Sascha Martens[1,2]

**Autophagy serves as a stress response pathway by mediating the degradation of cellular material within lysosomes. In autophagy, this material is encapsulated in double-membrane vesicles termed autophagosomes, which form from precursors referred to as phagophores. Phagophores grow by lipid influx from the endoplasmic reticulum into Atg9-positive compartments and local lipid synthesis provides lipids for their expansion. How phagophore nucleation and expansion are coordinated with lipid synthesis is unclear. Here, we show that Faa1, an enzyme activating fatty acids, is recruited to Atg9 vesicles by directly binding to negatively charged membranes with a preference for phosphoinositides such as PI3P and PI4P. We define the membrane-binding surface of Faa1 and show that its direct interaction with the membrane is required for its recruitment to phagophores. Furthermore, the physiological localization of Faa1 is key for its efficient catalysis and promotes phagophore expansion. Our results suggest a positive feedback loop coupling phagophore nucleation and expansion to lipid synthesis.**

## Introduction

Macroautophagy (hereafter autophagy) mediates the degradation of harmful material within the lysosomal system. It also protects cells during starvation by providing energy and building blocks for the synthesis of essential factors. Autophagy therefore represents a major pillar of the cellular stress response, and various diseases are associated with defects in this process (Levine and Kroemer, 2019; Mizushima and Komatsu, 2011). The hallmark of autophagy is the de novo formation of a double membrane organelle, the autophagosome. Upon induction of autophagy, the nucleation of a small membrane precursor referred to as phagophore is initiated. This membrane structure gradually captures cytoplasmic material as it grows. After the closure of the phagophore, the resulting autophagosome fuses with lysosomes (or the vacuole in yeast and plants) where the inner membrane and the cargo are degraded (Lamb et al., 2013; Wen and Klionsky, 2016).

The biogenesis of autophagosomes is mediated by a set of conserved factors referred to as the autophagy machinery (Chang et al., 2021; Mizushima et al., 2011; Nishimura and Tooze, 2020; Xie and Klionsky, 2007). Over the past years, a consensus model for how this machinery acts to mediate the formation and expansion of phagophores has emerged (Holzer et al., 2024). According to this model, phagophore formation is initiated in proximity to the endoplasmic reticulum (ER) by membrane precursors containing the lipid scramblase Atg9 (Axe et al.,

2008; Bieber et al., 2022; Gómez-Sánchez et al., 2018; Graef et al., 2013; Hayashi-Nishino et al., 2009; Suzuki et al., 2013; Yamamoto et al., 2012; Ylä-Anttila et al., 2009). The ER and the Atg9 positive membrane precursors are connected by the lipid transfer protein Atg2 (Chowdhury et al., 2018; Gómez-Sánchez et al., 2018; Graef et al., 2013; Kotani et al., 2018). The Atg1 kinase complex and the class III phosphatidylinositol 3-kinase complex 1 (PI3KC3C1) in conjunction with the PROPPINs facilitate the formation of these membrane contact sites and promote the subsequent expansion of the initial membrane seeds by recruiting and activating the Atg8 lipidation machinery (Dooley et al., 2014; Fracchiolla et al., 2020). This machinery in turn mediates the conjugation of Atg8 to phosphatidylethanolamine (PE) on the expanding phagophore (Cheong and Klionsky, 2008; Fujita et al., 2008; Hanada et al., 2007; Harada et al., 2019; Ichimura et al., 2000; Juris et al., 2015; Kihara et al., 2001; Obara et al., 2008; Papinski et al., 2014; Slobodkin and Elazar, 2013; Romanov et al., 2012; Suzuki et al., 2015; Yamamoto et al., 2016).

The main lipid source for autophagosome biogenesis is the ER from where the lipids are transported to the phagophore by Atg2 (Maeda et al., 2019; Osawa et al., 2019; Otomo and Maeda, 2019; Valverde et al., 2019). Another lipid transfer protein, Vps13, is partially redundant with Atg2 with respect to the lipid transfer activity and was also shown to directly interact with ATG9A in mammals (Dabrowski et al., 2023; van Vliet et al., 2024). In the

[1]Max Perutz Labs, Vienna BioCenter Campus (VBC), Vienna, Austria;   [2]Max Perutz Labs, Department of Biochemistry and Cell Biology, University of Vienna, Vienna, Austria;   [3]Vienna BioCenter PhD Program, A Doctoral School of the University of Vienna, Medical University of Vienna, Vienna, Austria.

*V. Baumann, S. Achleitner, and S. Tulli contributed equally to this paper.   Correspondence to Sascha Martens: sascha.martens@univie.ac.at.

phagophore, the ER-derived lipids are distributed between the two leaflets by the Atg9 lipid scramblase (Guardia et al., 2020; Maeda et al., 2020; Matoba et al., 2020). It has been shown that lipid synthesis in the ER is required to provide sufficient lipids for autophagosome biogenesis (Andrejeva et al., 2020; Polyansky et al., 2022; Schütter et al., 2020). In particular, de novo synthesized lipids in the ER can be detected in considerable amounts in autophagosomal membranes (Andrejeva et al., 2020; Orii et al., 2021; Schütter et al., 2020). In mammalian cells, autophagosome biogenesis takes place at ER sites enriched in phosphatidylinositol (PI) synthase (PIS) (Nishimura et al., 2017). In *Saccharomyces cerevisiae,* the long-chain fatty acid-coenzyme A (acyl-CoA) synthetases (ACSs), Faa1 and Faa4, which catalyze the first step in lipid synthesis by generating acyl-CoA, localize to phagophores. Their catalytic activities are redundant with each other and with the fatty acid synthase (FAS). Upon inhibition of FAS, the recruitment of Faa1 to phagophores as well as its catalytic activity becomes critical for autophagosome biogenesis. The recruitment of Faa1 to Atg8 positive structures at detectable levels depends on Atg1, Atg5, Atg9, and Atg14 (Schütter et al., 2020). Yet, the mechanistic details of Faa1 recruitment and activation on membranes remain elusive.

Here, we set out to characterize the membrane recruitment mechanism of Faa1 and its relevance for autophagosome formation. Our results suggest a positive feedback loop where the recruitment of Faa1 to Atg9 vesicles and phagophores is key for local lipid synthesis to sustain phagophore expansion. In turn, phagophore expansion requires phosphorylation of PI to phosphatidylinositol 3-phosphate (PI3P) by the PI3KC3C1 to continuously recruit Faa1.

## Results and discussion

### Faa1 is recruited to Atg9 vesicles by direct membrane binding

Faa1 accumulates on phagophores where it is proposed to locally activate fatty acids for lipid synthesis in the ER. The generated phospholipids are subsequently transported into the expanding phagophore by Atg2 (Fig. 1 A). Relocalization of Faa1 from the site of autophagosome biogenesis to the plasma membrane results in an impairment of autophagic flux, underlining the pivotal role of localized de novo lipid synthesis for phagophore expansion (Schütter et al., 2020). Interestingly, proteomic analyses have identified Faa1 in Atg9 vesicle–enriched fractions, suggesting an early role for Faa1 during autophagosome biogenesis (Sawa-Makarska et al., 2020). This is likely a conserved mechanism since analysis of the human ATG9A interactome also revealed several proteins from this fatty acid CoA ligase family (van Vliet et al., 2024). To obtain insights into the mechanistic basis of Faa1 recruitment to the site of autophagosome biogenesis, we tested if Faa1 could directly bind to Atg9 vesicles. We isolated Atg9 vesicles from yeast cells and tethered them via EGFP-tagged Atg9 to GFP-trap beads. Upon incubation with Faa1-mCherry purified from *S. cerevisiae*, we observed the binding of Faa1 to the vesicle-covered beads by confocal microscopy (Fig. 1 B). Atg9 vesicles have been shown to act as an assembly platform for the downstream autophagy machinery including the lipid

binding proteins Atg21 and Atg18 that directly bind PI3P generated by the PI3KC3C1 (Sawa-Makarska et al., 2020). Therefore, we tested whether the recruitment of Faa1 to Atg9 vesicles could be enhanced by active PI3KC3C1. However, a comparison of Faa1 binding in the presence of ATP/AMP-PNP and PI3KC3C1 revealed no difference under these conditions.

Yeast Atg9 vesicles have a very distinct lipid composition characterized by a high PI content of 44% (Sawa-Makarska et al., 2020). Aiming to distinguish if Faa1 recruitment to Atg9 vesicles occurs through direct membrane binding or rather relies on interactions with other proteins, we generated artificial proteoliposomes. These proteoliposomes mirror the native lipid composition of Atg9 vesicles + PI3P and contain Atg9 but lack additional proteins. As a control, proteoliposomes only consisting of the neutral lipid 1-palmitoyl-2-oleoylphosphatidylcholine (POPC) and Atg9-EGFP were used (Fig. 1 C). While we observed robust Faa1 recruitment to proteoliposomes with an Atg9 vesicle–like lipid composition, POPC Atg9-proteoliposomes showed barely any binding of Faa1. This indicates that Faa1 recruitment is mainly driven by its direct interaction with specific phospholipids and that Atg9 is not sufficient to recruit Faa1.

### Selective membrane recruitment of Faa1 occurs through interaction with negatively charged lipids including PI3P and PI4P

To determine how Faa1 localizes to membranes, we first analyzed which type of lipids are required for Faa1 recruitment by employing a microscopy-based membrane binding assay as well as a liposome cosedimentation assay (Fig. 2, A and B; and Fig. S1 A). First, we used liposomes mimicking an Atg9 vesicle lipid composition with and without PI3P. The presence of PI3P did not significantly alter the already robust binding of Atg9 vesicle–like liposomes to Faa1, possibly due to their native highly charged composition (44% PI). Therefore, we proceeded with a systematic reduction in the percentage of negatively charged PI/1-palmitoyl-2-oleoylphosphatidylserine (POPS) (Fig. 2, A and B; and Fig. S1 A). We observed a gradual decrease of membrane recruitment as the charge was reduced. Consistently, Faa1 binding to liposomes comprised solely of POPC was barely detectable.

The PI and POPS titration experiments (Fig. 2, A and B; and Fig. S1 A) revealed a steep decrease in membrane binding when the percentage of negatively charged lipids fell below 20–30%. This caught our attention because the ER, which is the major lipid donor for phagophore expansion, has a lower percentage of negatively charged lipids compared to the Atg9 vesicles (Sawa-Makarska et al., 2020; van Meer et al., 2008). To sustain Faa1 membrane binding during lipid influx from the ER, the local generation of negatively charged lipids such as PI3P might therefore be necessary. Consistently, Atg14 was shown to be required for the phagophore localization of Faa1 (Schütter et al., 2020). We therefore investigated whether the presence of PI3P or phosphatidylinositol 4-phosphate (PI4P) enhances Faa1 binding under these more stringent conditions by conducting experiments with liposomes characterized by the same net charge but with and without PI3P/PI4P. Intriguingly, under these conditions, a significant preference of Faa1 for membranes containing PI3P/PI4P was observed (Fig. 2 C and Fig. S1 B).

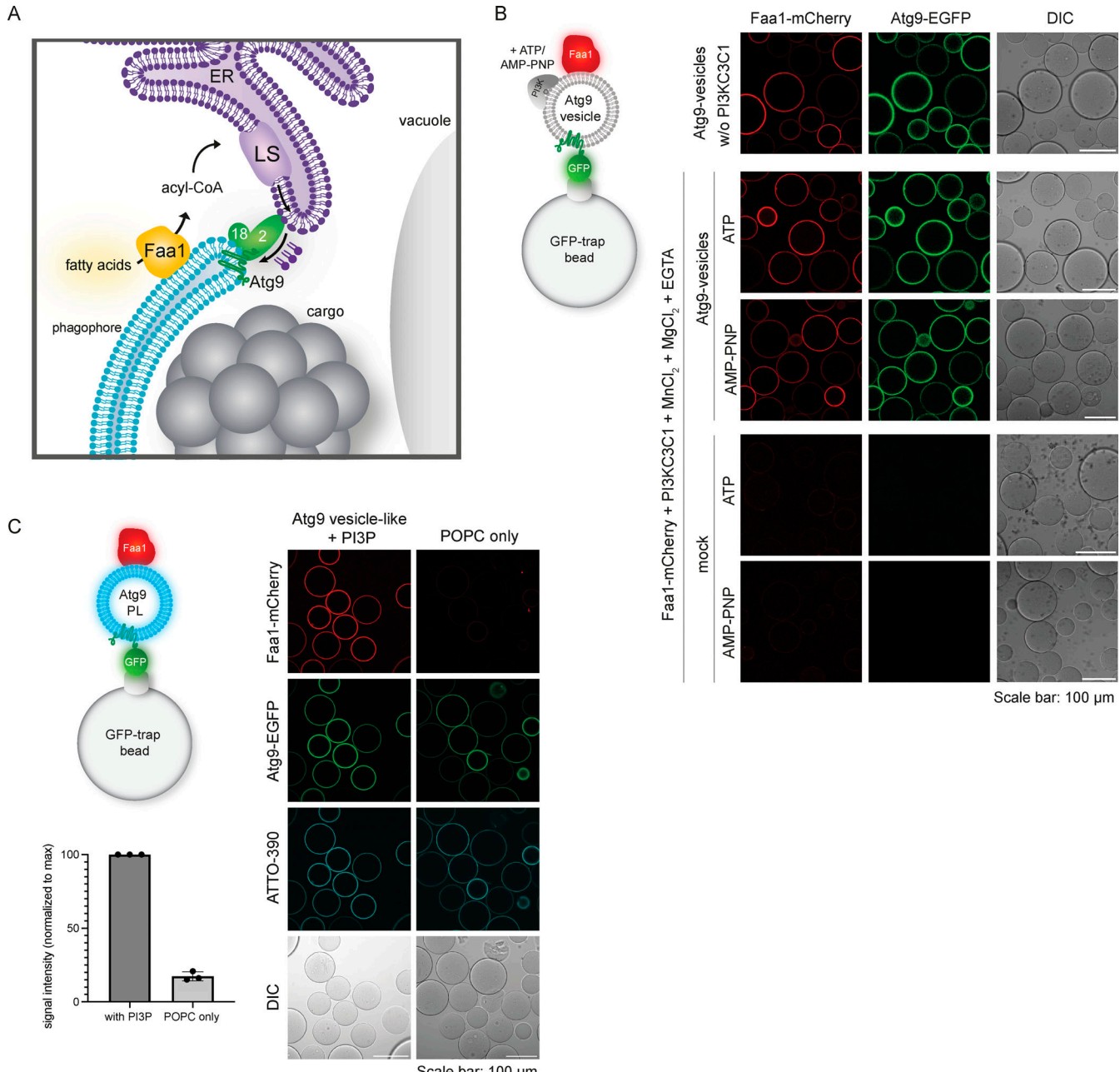

Figure 1. **Faa1 binds Atg9 vesicles as well as artificially generated Atg9 proteoliposomes. (A)** Model for phagophore expansion driven by localized de novo phospholipid synthesis. **(B)** Microscopy-based membrane binding assay testing the recruitment of Faa1-mCherry to isolated and immobilized endogenous Atg9 vesicles with and without active PI3KC3C1. The experimental setup is shown on the left. **(C)** Microscopy assay testing the recruitment of Faa1-mCherry to ATTO 390-1,2-dioleoylphosphatedylethanolamine (DOPE) labeled Atg9 proteoliposomes. Atg9-proteoliposomes were immobilized on GFP-trap beads. Quantification of Faa1 recruitment to Atg9 proteoliposomes can be seen on the bottom left. Data are means ± SD ($n$ = 3). The lipid composition of the Atg9 vesicle-like proteoliposomes is 42.5% POPC, 6% POPS, 6% POPE, 41.5% liver PI, 2.5% PI3P, and 1.5% ATTO 390-DOPE. PL = proteoliposome.

In cells, Faa1 localizes on growing phagophores and auto-phagosomes (Schütter et al., 2020) that in yeast range between 0.2 and 1 µm in diameter (Baba et al., 1994), while Atg9 vesicles have a diameter of 30–60 nm (Yamamoto et al., 2012). To identify any potential influence of membrane curvature, we tested Faa1 recruitment to giant unilamellar vesicles (GUVs). With a diameter >1 µm, GUVs can be considered a flat membrane surface. Faa1-mCherry robustly localized to 7-nitro-2-1,3-ben-zoxadiazol-4-yl (NBD)-labeled GUVs only in the presence of PI3P

(Fig. 2 D) as GUVs containing 27.5% POPS and no PI3P failed to bind Faa1. Interestingly, smaller liposomes with the same com-position were bound to Faa1 (Fig. 2 C). Nevertheless, a com-parison between differently sized small liposomes did not show a significant difference in binding to Faa1 (Fig. S1, C and D). Taken together, these results show that while membrane cur-vature might have a stimulatory effect on Faa1's membrane re-cruitment, the latter is predominantly triggered by negatively charged lipids. Therefore, we hypothesize that Faa1 is recruited

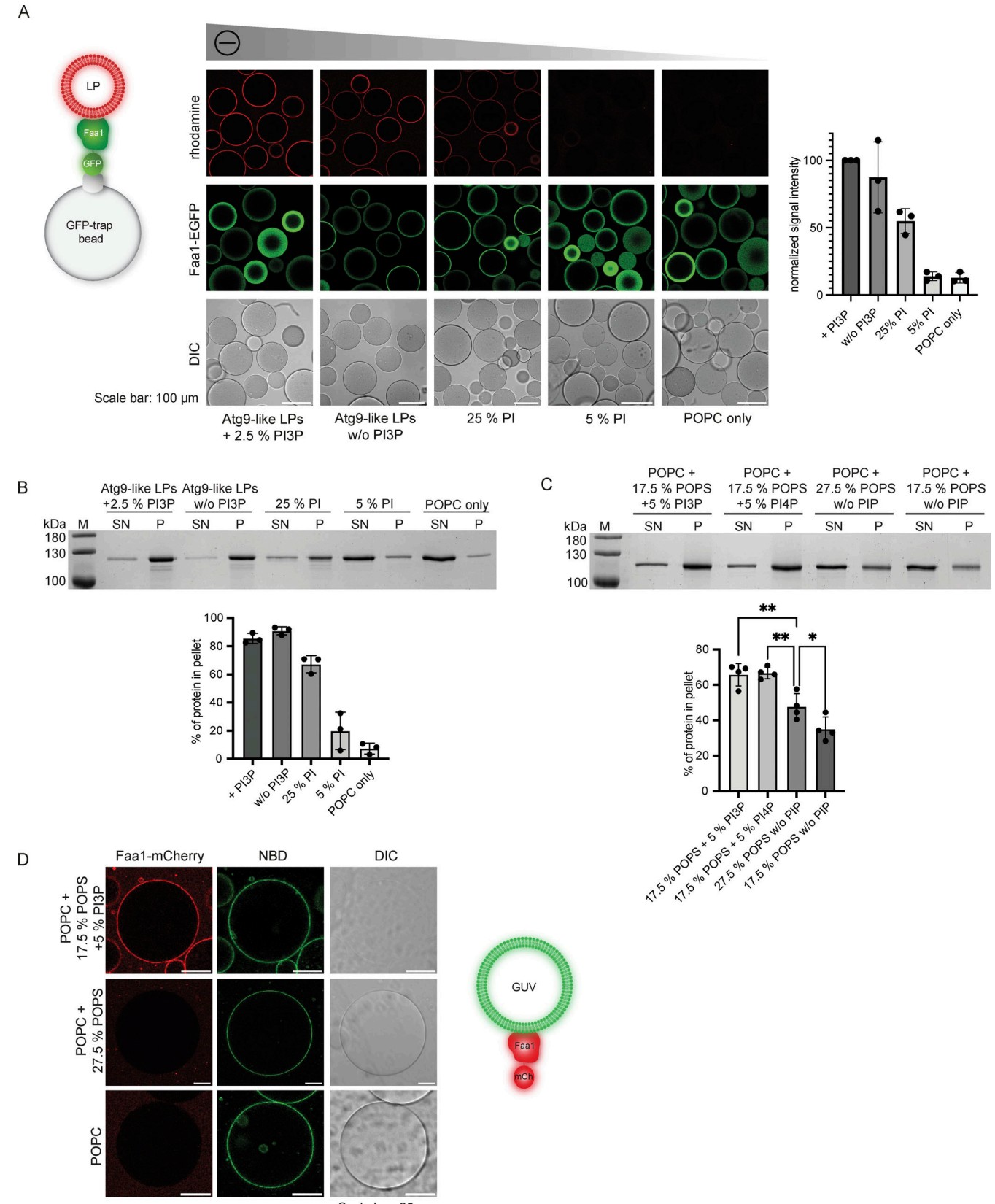

Figure 2. **Faa1 binds to negatively charged membranes with preference for PI3P/PI4P. (A)** Microscopy-based membrane binding assay showing that Faa1 is recruited to liposomes dependent on the net charge of the membranes. Faa1-EGFP was immobilized on GFP-trap beads and incubated with different lipid compositions. For visualization, they were supplemented with lissamine rhodamine-1,2-dihexadecanoylphosphatidylethanolamine (DHPE). Data are means ± SD ($n = 3$). LP = liposome. **(B)** Cosedimentation assay confirming the results from Fig. 2 A. Data are means ± SD ($n = 3$). LP = liposome, SN = supernatant,

P = pellet. Liposome compositions in 2A and 2B resemble Atg9 vesicles with 44% POPC, 6% POPS, 44% liver PI, 5.5% POPE, and 0.5% lissamine rhodamine-DHPE. For liposomes containing PI3P, the percentage of PI was reduced while a decreased percentage of PI was substituted with POPC. POPC-only liposomes contained 99.5% POPC and 0.5% lissamine rhodamine-DHPE. **(C)** Cosedimentation assay shows that Faa1 is preferentially recruited to liposomes containing phosphatidylinositides when compared to liposomes with the same net charge. All liposomes are composed of POPC with additionally the lipids indicated in the figures and 0.5% lissamine rhodamine-DHPE. Data are means ± SD ($n = 4$). One-way ANOVA: * < 0.05, ** < 0.01. SN = supernatant, P = pellet. **(D)** Microscopy-based GUV binding assay testing Faa1-mCherry recruitment to flat membranes. GUVs were composed of POPC with additionally the lipids indicated in the figures and 1.5% NBD-DPPE. Source data are available for this figure: SourceData F2.

to Atg9 vesicles by their high content of negatively charged PI at a very early stage of autophagy (Sawa-Makarska et al., 2020). As the Atg9 vesicles establish a connection with the less negatively charged ER via Atg2 and Vps13, these two lipid transfer proteins mediate the influx of phospholipids from the ER into the Atg9 vesicles (Dabrowski et al., 2023; Gómez-Sánchez et al., 2018; Graef et al., 2013; Kotani et al., 2018; van Meer et al., 2008). Under these more stringent conditions, continuous localization of Faa1 to the expanding vesicles likely depends on phosphorylation of PI to PI3P by the PI3KC3C1, consistent with the requirement for Atg14 for the localization of Faa1 on the phagophore (Schütter et al., 2020).

### A positively charged surface on Faa1 mediates membrane binding

The results above indicated that Faa1 directly binds negatively charged lipids. Such membrane-protein interactions are generally based on electrostatic forces between lipids and positively charged regions on the protein surface (Leonard et al., 2023). Since no experimental structure is available for Faa1 or any of its close homologs, we made use of the structure predicted by AlphaFold2 to identify the putative membrane binding site. The calculated electrostatic surface potential indeed suggested the presence of a positively charged surface on the protein (Fig. 3 A). Interestingly, this surface is also present in FACL4, the closest relative to Faa1 in humans (Fig. S2 A). Based on this hypothesis, we constructed a model illustrating the potential orientation of Faa1 on the membrane (Fig. 3 B). To test if this region is responsible for membrane binding, we mutated lysine residues located in the corresponding surface patch to aspartate. We generated a 4D mutant with K387, K388, K636, and K647, and a 6D mutant with K612 and K619 additionally mutated to aspartate. The 4D and 6D mutants behaved identically to the wild-type protein during the purification including size exclusion chromatography, where all proteins eluted as monomers, suggesting that the mutations did not result in protein misfolding and aggregation (Fig. S2 B). We then tested the mutants for their ability to bind to membranes in a liposome cosedimentation assay as well as a microscopy-based membrane protein interaction assay (Fig. 3, C and D). As expected, the wild-type protein robustly copelleted with Atg9 vesicle-like liposomes including PI3P. However, both mutants showed a significantly reduced cosedimentation with these liposomes (Fig. 3 C) as well as reduced recruitment under the microscope (Fig. 3 D), suggesting that the mutated surface is a major contributor to membrane binding.

After discovering the crucial role of the positively charged surface area of Faa1 in membrane binding in vitro, we further corroborated our findings in vivo. In yeast cells, Faa1 had previously been observed to localize to various cellular compartments, including the plasma membrane, ER, and autophagosomes (Schütter et al., 2020). To investigate the impact of mutating the identified positively charged region on the localization of Faa1, we employed genomically modified yeast strains. These strains coexpressed different variants of Faa1-3xGFP along with the mCherry-tagged autophagy marker Atg8. In line with previous observations, our live-cell imaging showed wild-type Faa1 being recruited to the plasma membrane, the ER, and Atg8-positive autophagosomal membranes upon autophagy induction with rapamycin (Fig. 3 E and Fig. S2 C). In contrast, both Faa1-4D and Faa1-6D did not bind membranes but remained cytosolic. Importantly, the expression levels of the mutants were comparable with those of the wild-type protein (Fig. S2 D). We conclude that a conserved positively charged surface on Faa1 mediates membrane binding to negatively charged membranes in vitro and in cells.

### Faa1 membrane binding enhances its enzymatic activity and thereby facilitates cell survival and autophagic flux

Next, we asked whether the interaction of Faa1 with membranes affects its enzymatic activity. To conveniently follow the activity of Faa1, we employed a coupled enzymatic assay (Ford and Way, 2015). This assay uses the AMP that is produced by Faa1 during the ATP-dependent acyl-CoA production for the oxidation of NADH to NAD$^+$. In contrast to NAD$^+$, NADH absorbs at 340 nm and can be measured spectroscopically (Fig. 4 A). While the positive control with AMP showed a rapid reduction of absorbance, the addition of Faa1 led only to a slow decrease in absorption (yellow curve in Fig. 4 B). Intriguingly, upon introduction of liposomes with an Atg9 vesicle-like composition with and without PI3P, we observed a substantial increase in Faa1's enzymatic activity that was not detectable with POPC liposomes. These results strongly imply that membrane binding is a crucial factor influencing Faa1 activity. Notably, this is not a general property of ACSs since FadD, a bacterial ACS derived from *E. coli*, is highly active in the absence of liposomes (Fig. S3 A). Next, we went on to test if the identified positively charged surface patch is required for the membrane-induced enhancement of Faa1 activity (Fig. 4 C). Indeed, neither Faa1-4D nor Faa1-6D exhibited the elevated enzymatic activity observed for the wild-type protein.

We subsequently investigated the potential impact of the membrane-binding mutants on autophagy in vivo. To this end, we coexpressed 2xGFP-Atg8 alongside either wild-type Faa1 or the mutant variants in starved *faa1Δfaa3Δfaa4Δ* cells. Autophagic flux was monitored utilizing the GFP-Atg8 assay, which exploits the distinct stabilities of Atg8 and GFP within the vacuole (Shintani and Klionsky, 2004). Under these conditions, we did not observe any difference in the autophagic flux between

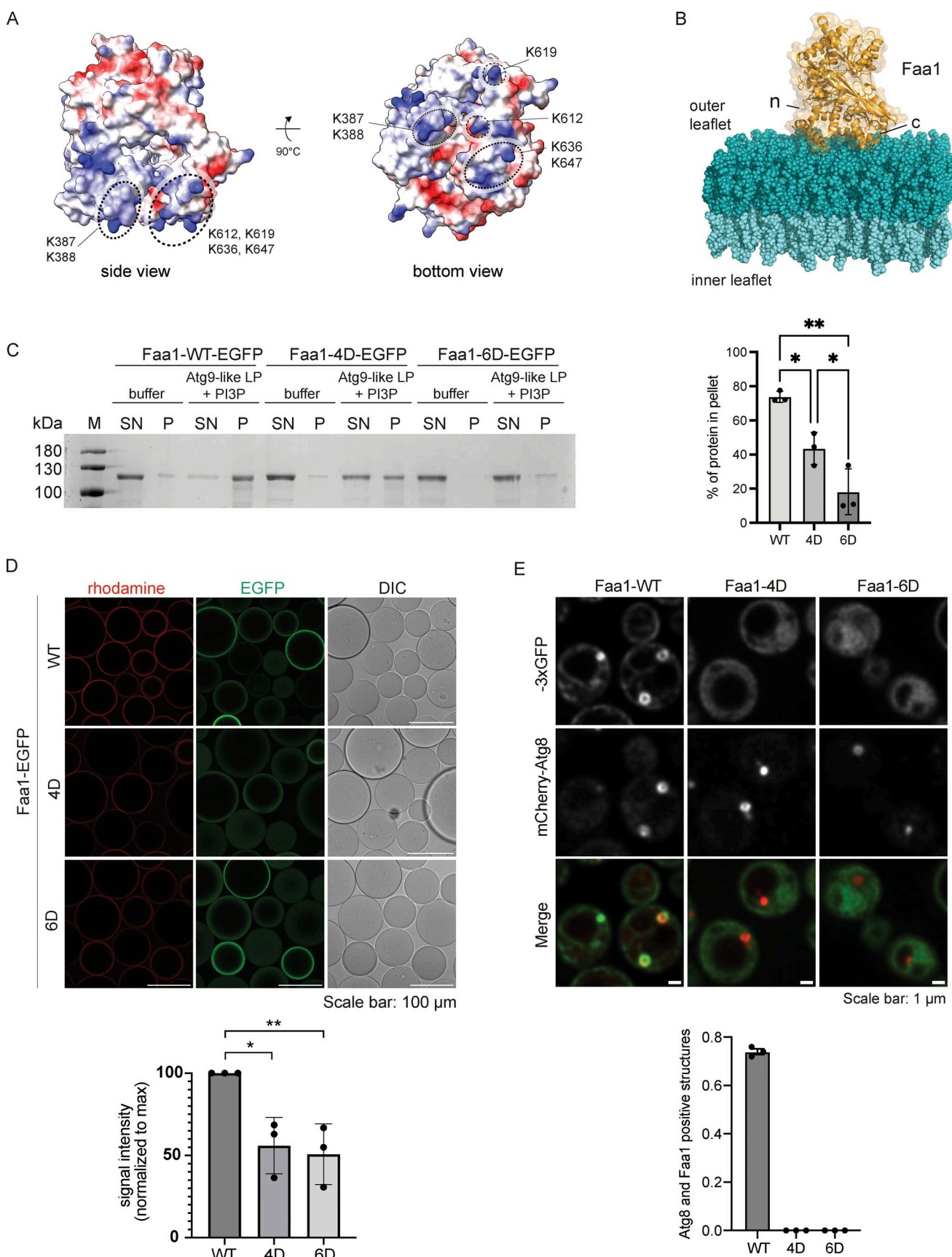

Figure 3. **A positively charged surface area mediates Faa1 membrane binding. (A)** Alphafold model of scFaa1. Electropositively and electronegatively charged areas are colored in blue and red, respectively. Neutral residues are in white. Mutated amino acids are highlighted with dotted circles. **(B)** Model of

Faa1 membrane interaction with Faa1 in yellow and the membrane bilayer in dark and light teal. The model was generated with CHARMM-GUI HMMM Builder (Jo et al., 2008; Qi et al., 2015). (n) refers to the N-terminus and (c) to the C-terminus of Faa1. **(C)** Left, cosedimentation assay showing that mutations of positively charged residues in the plane area (Faa1-4D, Faa1-6D) significantly reduce membrane binding compared with Faa1-WT. Right, respective quantification. Data are means ± SD ($n$ = 3). One-way ANOVA: * < 0.05, ** < 0.01. LP = liposome, SN = supernatant, P = pellet. **(D)** Microscopy-based membrane recruitment assay with different versions of Faa1. Faa1 is immobilized on GFP-trap beads, incubated with 1 mM of rhodamine labelled liposomes and imaged by microscopy. Liposome compositions for C and D resemble Atg9 vesicles + PI3P (44% POPC, 6% POPS, 41.5% liver PI, 5.5% POPE, 2.5% PI3P and 0.5% lissamine rhodamine-DHPE). Data are means ± SD ($n$ = 3). Two-sided t test: * < 0.05, ** < 0.01. **(E)** Top, Faa1-WT, Faa1-4D, and Faa1-6D localization in cells coexpressing mCherry-Atg8. Indicated strains were imaged via fluorescence microscopy after rapamycin treatment (1 h), scale bars, 1 µm. Bottom, respective quantification of Atg8 structures positive for Faa1 ($n$ = 3). Source data are available for this figure: SourceData F3.

Faa1 wild-type or the mutants (Fig. S3 B). Notably, in yeast cells, the activation of fatty acids can be carried out by both ACSs and FAS via two parallel pathways (Black and DiRusso, 2007). ACSs activity is key for the survival of starving cells upon FAS inhibition via cerulenin. We therefore examined whether Faa1-WT, Faa1-4D, and Faa1-6D could restore the viability of starved and cerulenin-treated *faa1Δfaa3Δfaa4Δ* cells. As expected, wild-type Faa1 expression fully rescued cell survival. However, cells expressing Faa1-4D or Faa1-6D were unable to sustain cell viability under these conditions (Fig. 4 D).

We then proceeded to evaluate the effect of the membrane-binding mutants on autophagosome biogenesis and autophagic flux upon FAS inhibition. Autophagosome formation was monitored by live cell imaging of 2xGFP-Atg8 (Fig. 4 E). Cells expressing Faa1-4D or Faa1-6D hardly showed any autophagosomes, suggesting severely impaired autophagy. To further corroborate these findings, we monitored autophagic flux by employing the GFP-Atg8 assay. As expected, we observed efficient GFP-Atg8 cleavage in cells expressing wild-type Faa1 (Fig. 4 F). In contrast, no free GFP could be detected in cells expressing Faa1-4D or Faa1-6D.

As an alternative and physiologically relevant way to inhibit FAS, we lowered the glucose concentration (0.01%) during starvation and assessed Faa1-4D and Faa1-6D function (Fig. S3 C). Under this metabolic condition, the effect of the Faa1-4D and Faa1-6D mutations was less prominent compared to the response seen with cerulenin treatment. However, when cells were imaged to monitor autophagosome biogenesis after 1 h of starvation in reduced glucose, Faa1-4D and Faa1-6D cells showed fewer autophagosome than wild-type Faa1. Interestingly, a reversal in this trend was observed after 1 h and 15 min with Faa1-4D and Faa1-6D cells showing more autophagosomes than wild-type Faa1 cells. This suggests that impaired Faa1 function combined with reduced glucose levels during starvation results in slower autophagosome biogenesis. Since the reduction of glucose during starvation did not entirely suppress autophagosome formation, we proceeded to assess autophagic flux. Further analysis showed a mild reduction in autophagic flux for the mutants compared with the wild-type protein (Fig. S3 D).

## Physiological Faa1 membrane localization is required for efficient lipid incorporation and maintenance of autophagic flux

We demonstrated that Faa1's enzymatic activity is substantially dependent on its localization and direct membrane binding via the identified positive surface patch. However, why this interaction is so important for its enzymatic activity is unclear. A plausible explanation is that the tight connection between Faa1

and the membrane aids the release of its product, acyl-CoA, into the lipid bilayer. To test this, we carried out a Förster Resonance Energy Transfer (FRET)-based assay. Liposomes containing two fluorophores were mixed with Faa1. The incorporation of acyl-CoA into the lipid bilayer would increase the distance between the two fluorescent lipids and result in an increase in NBD fluorescence. Indeed, upon the addition of CoA and subsequent synthesis of acyl-CoA, the NBD fluorescence increases for Faa1-WT (Fig. 5 A). Faa1-4D exhibits a diminished dequenching of the NBD-DPPE signal, indicating a reduced production and incorporation of acyl-CoA.

Next, we asked whether artificial tethering of Faa1-WT and Faa1-4D to POPC or Atg9-like liposomes would restore its activity. To assess this, we made use of 10xHis-tagged Faa1 and liposomes containing 1,2-dioleoyl-sn-glycero-3-[(N-(5-amino-1-carboxypentyl) iminodiacetic acid) succinyl] (DGS-NTA). Measurement of the enzymatic activity revealed that neither Faa1-WT tethering to POPC liposomes nor Faa1-4D tethering to Atg9 vesicle-like liposomes resulted in an enhanced catalytic function compared with liposomes without DGS-NTA (Fig. 5 B and Fig. S3 E). This suggests that the correct orientation of Faa1 on the membrane via the positively charged membrane binding patch is essential for its activity.

We proceeded to test if forcing Faa1-4D localization to membranes in vivo restores autophagy. To this end, we employed the PI3P-sensitive FYVE domain to recruit Faa1-WT and Faa1-4D to PI3P-rich membranes. Indeed, both Faa1-WT and Faa1-4D localized on Atg8 positive structures as well as on the vacuolar membrane in *faa1Δfaa3Δfaa4Δ* cells starved and treated with cerulenin (Fig. 5 C, left). When cell survival was assessed in this metabolic condition, only a very mild amelioration was observed comparing Faa1-4D-FYVE to Faa1-4D (Fig. 5 C, right). Consistently, while both Faa1-WT and Faa1-WT-FYVE showed vacuolar signal after 6 h of nitrogen starvation and cerulenin treatment, no signal was detectable in cells expressing Faa1-4D or Faa1-4D-FYVE (Fig. 5 D). Western blot analysis of mCherry-Atg8 cleavage confirmed a block of autophagy in Faa1-4D and Faa1-4D-FYVE strains (Fig. 5 E). Despite similar protein expression levels (Fig. S3 F), Faa1-WT-FYVE showed reduced flux compared with Faa1-WT (Fig. 5 E). The dependence of membrane binding for efficient Faa1 activity offers another layer of regulation to initiate lipid synthesis at the right time and space. In autophagy, this dependence allows positive feedback where the initial recruitment of Faa1 to Atg9-positive phagophore seeds initiates lipid synthesis (Fig. 5 F). The synthesized lipids support membrane expansion, creating additional binding sites for Faa1 and thus more substrates for lipid synthesis.

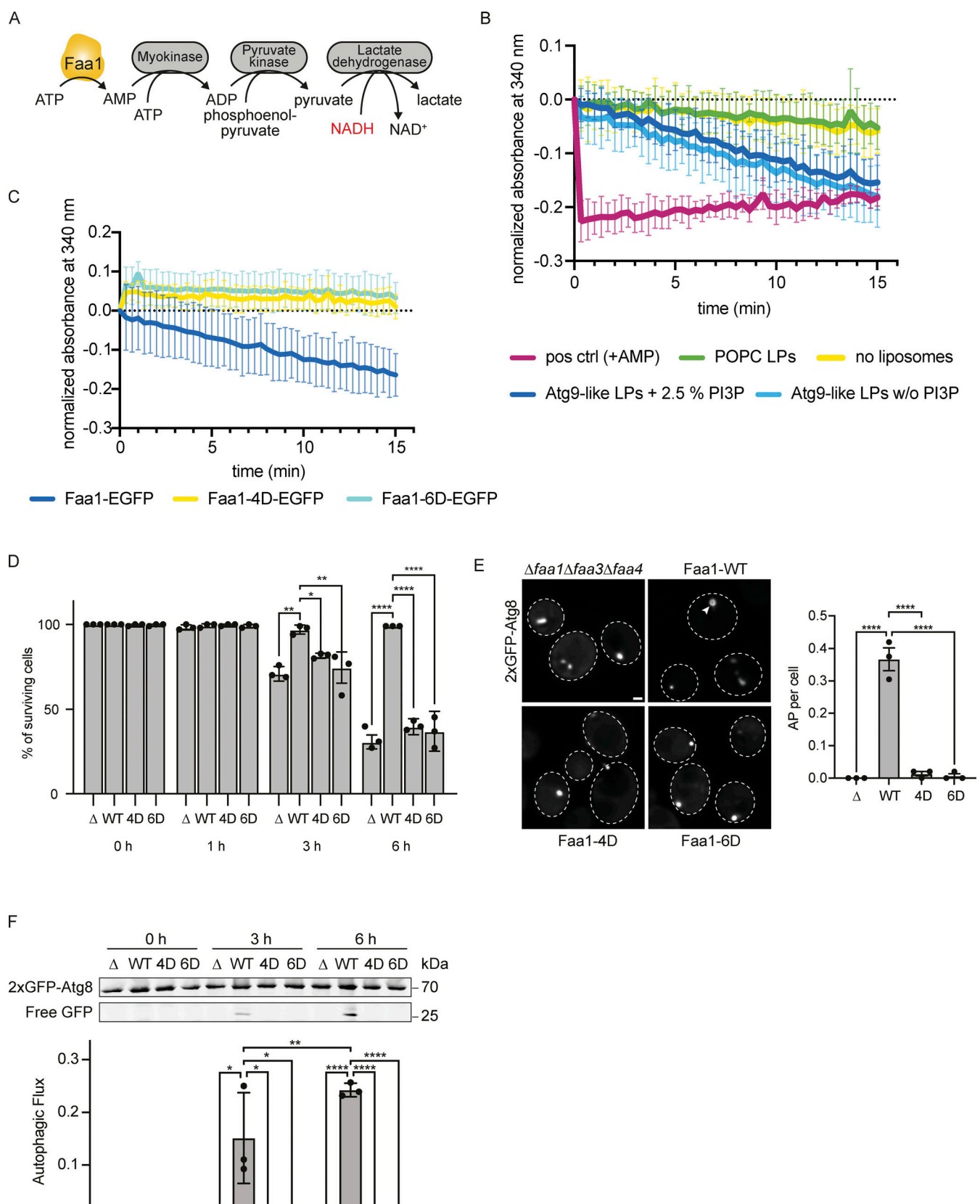

**Figure 4. Membrane binding of Faa1 is important for its activity. (A)** Scheme of the coupled enzymatic assay for testing the activity of Faa1. **(B)** Coupled enzymatic assay following the activity of Faa1 in the presence of different liposomes by measuring the absorbance of NADH at 340 nm. Liposome composition resembles Atg9 vesicles with 44% POPC, 6% POPS, 41.5% liver PI, 6% POPE, and 2.5% PI3P, whereas PI3P was substituted for liver PI in "Atg9-like LPs w/o

PI3P" and for "POPC only" liposomes contained 100% POPC. Data are means ± SD ($n$ = 3). The dotted line serves as a reference for 0. LP = liposome. **(C)** Enzymatic assay comparing Faa1-WT to Faa1-4D and Faa1-6D. Data are means ± SD ($n$ = 3). Liposome composition resembles Atg9 vesicles + PI3P (44% POPC, 6% POPS, 41.5% liver PI, 6% POPE, 2.5% PI3P). The dotted line serves as a reference for 0. **(D)** Survival of indicated strains during starvation + cerulenin after 0, 1, 3, and 6 h of treatment. Data are means ± SD ($n$ = 3; 100 cells/strain per replicate). One-way ANOVA: * < 0.05, ** < 0.01, *** < 0.001, **** < 0.0001. **(E)** Fluorescent microscopy assessment of autophagosome biogenesis in indicated strains after 1 h nitrogen starvation + cerulenin. Left: Representative images of the analyzed strains. Dashed lines indicate cell boundaries, arrowheads autophagosomes (AP). Scale bar, 1 µm. Right: Respective quantification of autophagosomes. Data are means ± SEM ($n$ = 3; 50 cells/strain per replicate). One-way ANOVA: **** < 0.0001. **(F)** Autophagic flux of indicated strains expressing 2xGFP-ATG8 after 0, 3, and 6 h of starvation + cerulenin. Data are means ± SD ($n$ = 3). One-way ANOVA: * < 0.05, ** < 0.01, **** < 0.0001. Source data are available for this figure: SourceData F4.

## Materials and methods

### Protein expression and purification

Yeast strains and constructs for protein expression can be found in Table S1 and Table S2, respectively. The purification procedures of proteins from these strains and constructs are described below.

Faa1-mCherry, Faa1-4D-mCherry, Faa1-EGFP, Faa1-4D-EGFP, Faa1-6D-EGFP, Faa1-10xHis, and Faa1-4D-10xHis were purified from the SMY464, SMY518, SMY465, SMY503, SMY506, SMY523, and SMY520 strains, respectively. Cells were grown at 30°C in YPG up to an $OD_{600}$ of 7–11. Cells were pelleted, washed once with cold $H_2O$, once with lysis buffer A (500 mM NaCl, 50 mM Hepes pH 7.5, 2 mM $MgCl_2$), and finally resuspended in lysis buffer containing cOmplete protease inhibitors (5056489001; Roche), a protease inhibitor mix FY (39104; Serva), DNAse I (DN25; Sigma-Aldrich), benzonase (E1014; Sigma-Aldrich), and 1 mM DTT. Resuspended cells were frozen in liquid nitrogen as pearls and lysed via freezer milling. The milled powder was thawed and resuspended in lysis buffer A by rolling gently at 4°C. Lysates were cleared by centrifugation (46,000 × $g$ for 45 min at 4°C in a Beckman Ti45 rotor). The supernatant was incubated with IgG Sepharose 6 Fast Flow beads (17096901; Cytiva) on a rotary wheel for 1 h at 4°C. Beads with bound protein were washed twice with lysis buffer A, once with high salt buffer (700 mM NaCl, 25 mM Hepes pH 7.5, 1 mM DTT), and twice with lysis buffer A without protease inhibitors. Bound protein was cut off the beads with TEV protease for 1 h at 16°C. The cleaved protein was concentrated and loaded onto a Superose 6 Increase 10/300 size exclusion chromatography column (29091596; Cytiva). Elution was carried out with 150 mM NaCl, 25 mM Hepes pH 7.5, and 1 mM DTT buffer. Fractions containing the protein of interest were pooled, concentrated, frozen in liquid nitrogen, and stored at –70°C.

Expression and purification of the PI3KC3C1 and Atg9-EGFP were carried out as described in Sawa-Makarska et al. (2020).

PI3KC3C1 subunits (Vps15, Vps34, Atg6, and Atg14) were coexpressed in baculovirus-infected Sf9 insect cells via SMC1181 containing the codon-optimized ORFs. Atg9-EGFP was also expressed in baculovirus-infected Sf9 insect cells via SMC1230 containing the codon-optimized ORF. Sf9 cells were infected with 1 ml P1 virus for PI3KC3C1 or 10 ml P2 virus for Atg9 per 1 liter medium supplemented with penicillin and streptomycin containing $10^6$ Sf9 cells/ml. Cells were harvested at 715 × $g$ for 15 min, washed with 30 ml of PBS, and frozen. For the purification of PI3KC3C1, the cell pellets were lysed in lysis buffer containing 300 mM NaCl, 50 mM Tris pH 8.8, 1 mM DTT, 0.5% 3-[(3-cholamidopropyl) dimethylammonio]-1-propanesulfonate (CHAPS) (D99009; Glycon), benzonase (E1014; Sigma-Aldrich), protease inhibitor cocktail (P8849; Sigma-Aldrich), and cOmplete protease inhibitor cocktail (5056489001; Roche) using a tissue homogenizer. The lysate was cleared by centrifugation (16,000 × $g$ for 40 min at 4°C). The supernatant was incubated with IgG Sepharose 6 Fast Flow beads (17096901; Cytiva) and incubated for 1 h at 4°C rotating. Beads with bound protein were washed twice with buffer A (300 mM NaCl, 50 mM Tris pH 8.0, 1 mM DTT, 0.5% CHAPS [D99009; Glycon]) and twice with buffer A without CHAPS. The bound protein was cleaved off the beads by TEV protease in buffer a without CHAPS overnight at 4°C. The eluted PI3KC3C1 was concentrated and loaded onto a Superdex S200 Increase 10/300 column (28990944; Cytiva). Elution was carried out with 20 mM Tris pH 8.0, 150 mM NaCl, 20 mM Tris pH 8.0, and 1 mM DTT. Fractions containing the complex were collected, concentrated, flash frozen, and stored at –70°C.

For the purification of Atg9-EGFP, the cell pellet was lysed in lysis buffer containing 500 mM NaCl, 50 mM HEPES pH 7.5, 10% glycerol (G5516; Sigma-Aldrich), 2 mM b-mercaptoethanol (8.05740; Sigma-Aldrich), 1 mM $MgCl_2$, benzonase (E1014; Sigma-Aldrich), and cOmplete protease inhibitor cocktail (5056489001; Roche) using a tissue homogenizer. The lysate was cleared by centrifugation (5,600 × $g$, 15 min, 4°C). The membranes were pelleted by centrifugation of the cleared lysate at 185,000 × $g$ for 1 h and resuspended for 2 h at 4°C in lysis buffer containing 2% n-dodecyl-β-D-maltoside (DDM) (D97002; Glycon). The insoluble material was removed by centrifugation (185,000 × $g$, 1 h). The supernatant was loaded onto a 1 ml His-Trap FF column (17531901; Cytiva) from which it was eluted with a stepwise gradient of imidazole, from 0 to 300 mM imidazole in 500 mM NaCl, 50 mM HEPES pH 7.5, 10% glycerol (G5516; Sigma-Aldrich), and 0.3% DDM (D97002; Glycon). The fractions containing the protein of interest were pooled, concentrated, and loaded onto a Superose 6 10/300 column (17517201; Cytiva). Elution was carried out with 500 mM NaCl, 50 mM HEPES pH 7.5, 10% glycerol (G5516; Sigma-Aldrich), and 0.2% DDM (D97002; Glycon). The fractions containing the protein were incubated with 150 µl of NiNTA Agarose beads (745400; Macherey-Nagel) for 3 h at 4°C. The beads were washed with 300 mM NaCl, 25 mM Tris pH 7.4, and 0.04% DDM (D97002; Glycon) before the protein was eluted with buffer supplemented with 300 mM imidazole. The imidazole was removed via dialysis overnight at 4°C against 25 mM Tris pH 7.4, 300 mM NaCl, and 0.04% DDM (D97002; Glycon).

The protein FadD was overexpressed from the construct SMC1496 in *E. coli* Rosetta pLysS. Cells were grown in lysogeny

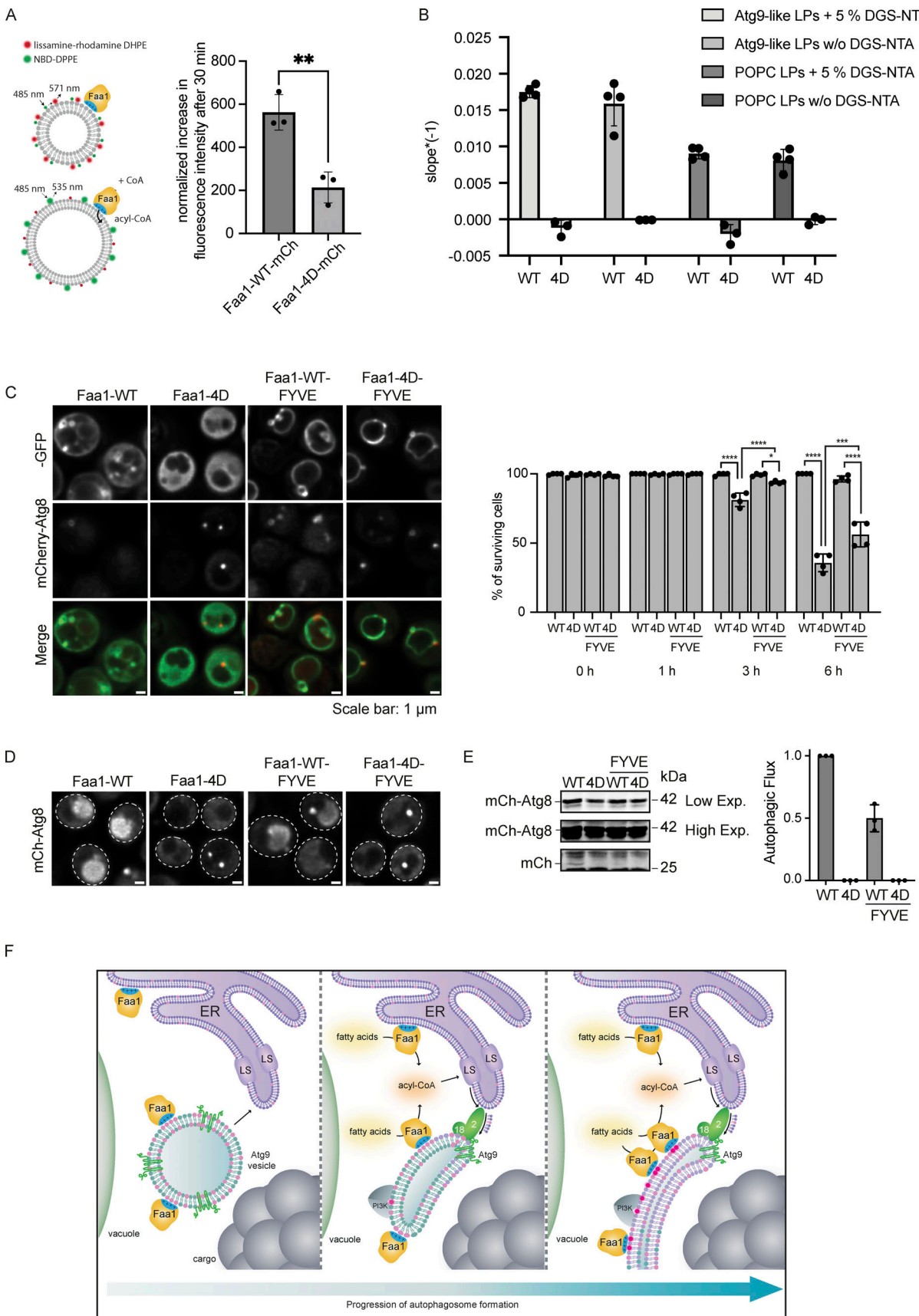

Figure 5. **Physiological Faa1 membrane localization is required for efficient lipid incorporation and maintenance of autophagic flux. (A)** Quantification of a FRET assay measuring the incorporation of acyl-CoA into the membrane. The normalized difference in fluorescence intensity between activated (+CoA)

and not activated (–CoA) Faa1-WT and Faa1-4D after 30 min is depicted. Liposome composition resembled Atg9 vesicles with 44% POPC, 6% POPS, 2% POPE, 44% liver PI, 2% NBD-DPPE, and 2% lissamine rhodamine-DHPE. Data are means ± SD ($n$ = 3). Two-sided $t$ test: ** < 0.01. **(B)** The maximum reaction rate of Faa1-WT-10xHis and Faa1-4D-10xHis when incubated with Atg9-like liposomes or POPC liposomes with or without 5% DGS-NTA was measured with a coupled enzymatic assay. Liposome composition resembled Atg9 vesicles ("Atg9-like LPs") with 44% POPC, 6% POPS, 41.5% liver PI, 6% POPE, and 2.5% PI3P or only contained 100% POPC ("POPC LPs"). For liposomes containing 5% DGS-NTA the percentage of POPC was reduced. Data are means ± SD ($n \geq$ 3). LP = liposome. **(C)** Left: Localization of Faa1-WT, Faa1-4D, Faa1-WT-FYVE, and Faa1-4D-FYVE in cells coexpressing mCherry-ATG8. Indicated strains were imaged via fluorescence microscopy after starvation + cerulenin treatment (3 h), scale bars, 1 µm. Right: Quantification of survival of indicated strains during starvation + cerulenin after 0, 1, 3, and 6 h of treatment. Data are means ± SD ($n$ = 4; 100 cells/strain per replicate). One-way ANOVA: * < 0.05, **** < 0.0001. **(D)** Representative images of the vacuolar signal of indicated strains after 6 h of starvation + cerulenin. **(E)** Autophagic flux of indicated strains expressing mCherry-ATG8 after 6 h of starvation + cerulenin. Data are means ± SD ($n$ = 3). **(F)** Model for the role of Faa1 during autophagosome formation. Source data are available for this figure: SourceData F5.

broth medium at 37°C. At an $OD_{600}$ of 0.9, the protein over-expression was induced with 0.5 mM IPTG. After 2.5 h, the cells were harvested by centrifugation, resuspended in lysis buffer (500 mM NaCl, 50 mM HEPES pH 7.4, 2 mM MgCl2, 2 mM b-mercaptoethanol [8.05740, Sigma-Aldrich]), and lysed with a cell disruptor. The lysate was cleared at 52,000 × $g$ for 40 min at 4°C, and the supernatant was loaded onto a 1 mL HisTrap HP column (17524701, Cytiva), from which it was eluted with a stepwise gradient of imidazole, from 0 to 300 mM imidazole in 500 mM NaCl, 50 mM HEPES pH 7.4. Fractions containing the protein were collected, concentrated, and loaded onto a Superdex 200 16/60 column (28989335, Cytiva). The protein was eluted with 500 mM NaCl, 50 mM HEPES pH 7.4. Fractions containing the protein were pooled, concentrated, and stored at –70°C.

### Atg9 vesicle isolation
Atg9 vesicle isolation was mainly carried out according to Sawa-Makarska et al. (2020) with minor optimization steps. Briefly, binding of the cleared Atg9-EGFP vesicles to the IgG Sepharose 6 Fast Flow beads (17096901; Cytiva) was carried out o/n. Cleavage from the beads was carried out for 3 h at 4°C. The final reaction buffer was changed to 135 mM NaCl, 25 mM Tris pH 7.4.

For the microscopy-based interaction assay between Atg9 vesicles and Faa1, bead-bound Atg9 vesicles were incubated for 2 h at RT with and without 50 nM of PI3KC3C1 including 0.5 mM $MgCl_2$, 2 mM $MnCl_2$, 1 mM EGTA (E4378; Sigma-Aldrich), and 0.2 mM ATP/AMP-PNP (A2383/A2647; Sigma-Aldrich). Next, 1 µM of Faa1-mCherry was added and incubated for 2 h. Imaging was carried out as described for the microscopy-based protein-membrane interaction assays.

### Formation of liposomes
Liposomes were prepared following the standard procedure as described in Sawa-Makarska et al. (2020). In brief, lipids were mixed in a glass vial and dried under an argon stream. The lipids were then dried further 1 h under vacuum. After rehydration in 150 mM NaCl, 25 mM Tris pH 7.4 buffer, the mixes were gently resuspended and sonicated for 2 min. The formed liposomes were then extruded first through a 0.4-µm and then a 0.1-µm membrane (10419504, 10417104; Whatman) using the Mini Extruder from Avanti Polar Lipids Inc. For experiments to determine curvature dependency, liposomes were either extruded through only a 0.4-µm filter or through a 0.4-µm and additionally a 0.05-µm filter (800308; Whatman). The final liposome suspensions have a concentration of 1 or 2 mM.

### Formation of Atg9-proteliposomes
Proteoliposomes formation was performed as described in Sawa-Makarska et al. (2020). In brief, liposomes were incubated for 1 h at room temperature with CHAPS (D99009; Glycon) at a concentration of 2.5%. The solution was then mixed with the same volume of a 1 µM Atg9-EGFP solution and incubated at room temperature for 1.5 h. The mix was diluted with 300 mM NaCl, 25 mM Tris pH 7.4 by a factor of 10, and further dialyzed against 300 mM NaCl, 25 mM Tris pH 7.4 overnight at 4°C. Bio-Beads SM2 (1523920; Bio-Rad Laboratories) were added to the proteoliposomes and incubated for 1 h at room temperature. The Bio-Beads were removed, and insoluble protein was pelleted at 20,000 × $g$ for 30 min. The supernatant contained the proteoliposomes that were used for the experiment.

### Formation of GUVs
GUV preparation was carried out by electroformation. 3 µl of the respective liposome mixture were spotted onto an indium-tin-oxide-coated glass slides in a dropwise manner. After desiccation for at least 3 h under vacuum, the electroformation chambers were assembled by using silicon isolators and copper tape. The chambers were filled with 300 mM sucrose solution. The electroformation protocol contained three phases. Phase 1: 30 min. Sine wave with frequency $f$ = 10 Hz. Peak-to-peak amplitude ramps up linearly from 0.05 to 1.41 V. Phase 2: 120 min. Sine wave with frequency $f$ = 10 Hz. Peak-to-peak amplitude held constant at 1.41 V. Phase 3: 30 min. Square wave with frequency $f$ = 4.5 Hz. Peak-to-peak amplitude held constant at 2.12 V. Electroformation was conducted at 60°C. Formed GUVs were diluted 1:3 in GUV dilution buffer (135 mM NaCl, 20 mM Hepes pH 7.5). Before the GUVs and proteins were pipetted onto the plate, the wells were blocked with a blocking solution (2.5 mg/ml BSA [A9647; Sigma-Aldrich] in 50 mM Tris-HCl pH 7.4, 150 mM NaCl) for 1 h and washed twice with the reaction buffer.

### Membrane recruitment—GUV assays
For Faa1 recruitment, three different GUV mixes were prepared: 99.5% POPC (850457C; Avanti Polar Lipids, Inc.) and 0.5% NBD-DPPE (810144C; Avanti Polar Lipids, Inc.) for POPC only, 77% POPC (850457C; Avanti Polar Lipids, Inc.), 17.5% POPS (840034C; Avanti Polar Lipids, Inc.), 5% PI3P (850150P; Avanti Polar Lipids, Inc.), and 0.5% NBD-DPPE (810144C; Avanti Polar Lipids, Inc.) as well as 72% POPC (850457C; Avanti Polar Lipids, Inc.), 27.5% POPS (840034C; Avanti Polar Lipids, Inc.) and 0.5% NBD-DPPE (810144C; Avanti Polar Lipids, Inc.). 30 µl of the GUV

mixture were pipetted to a 96-well glass-bottom microplate (655892; Greiner Bio-One). Faa1 was added to a final concentration of 1 µM in a final reaction volume of 35 µl reaction buffer (150 mM NaCl, 25 mM Tris-HCl pH 7.4). After 1 h of incubation time, GUVs were imaged at room temperature with a Zeiss LSM700 confocal microscope (RRID:SCR_017377) and a Plan-Apochromat 20×/0.8 WD 0.55 mm objective. Imaging software was ZEN 2012 SP5 (RRID:SCR_013672) running on Windows 10 (64-bit).

### Microscopy-based protein–membrane interaction assays

For Fig. 2 A, Fig. 3 D, and Fig. S1, A and B, wild type or mutant versions of Faa1-EGFP were recruited to GFP-Trap Agarose beads (gta, Chromotek). Assays were performed under equilibrium conditions with 1 mM of the prey liposomes in buffer (150 mM NaCl, 50 mM Hepes/Tris pH 7.5) labeled with 0.5% lissamine rhodamine-DHPE (L-1392; Invitrogen). Beads were imaged at room temperature with a Zeiss LSM700 confocal microscope (RRID:SCR_017377) and a Plan-Apochromat 20×/0.8 WD 0.55 mm objective. The imaging software was Zeiss ZEN 2012 SP5 (RRID:SCR_013672) running on Windows 10 (64-bit). For quantification, lines were drawn across each bead in Fiji (Version 2.9.0/1.53t, RRID:SCR_002285), and the maximal values across the lines were taken. Values were averaged for each sample within each replicate and then among replicates. The final values were then normalized to max. Beads from three independent experiments were quantified for the rhodamine signal.

For testing the recruitment of Faa1 to Atg9 proteoliposomes, Atg9-EGFP proteoliposomes were immobilized on GFP-Trap Agarose beads (gta, Chromotek). The assay was performed under equilibrium conditions with a prey concentration of 1 µM Faa1-mCherry. Imaging and analysis were carried out as stated above for the membrane–protein interaction assay.

### Liposome cosedimentation assay

5 µg of the Faa1 variants were incubated with 25 µl (50 µl for Fig. 3 C) liposomes for 30 min at room temperature. The reaction was centrifuged at 200,000 × g for 15 min at 22°C (TLA100 rotor; Beckman-Coulter Optima MAX-XP). The supernatant and pellet were loaded separately on an SDS/polyacrylamide gel in equal amounts. The quantification of the bands was performed with the Analyze gels tool of Fiji (Version 2.9.0/1.53t, RRID:SCR_002285). The percentage of pelleted protein was calculated and the buffer control was subtracted.

### Faa1 activity assay

The activity of the Faa1 variants was determined by measuring the production of AMP with an adapted protocol from Ford and Way (2015). 5 µg (6 µg for Fig. 4 B) of the Faa1 variants were incubated with 20 µl (24 µl for Fig. 4 B) of 1 mM liposomes for 30 min at room temperature. The reaction was mixed in a volume of 100 µl containing the Faa1 liposomes mix, 8 mM $MgCl_2$, 2 mM EDTA, 0.1 mM oleic acid (O1008; Sigma-Aldrich), 0.2 mM NADH (N8129; Sigma-Aldrich), 0.3 mM phosphoenolpyruvate (860077; Sigma-Aldrich), 0.1 mM BSA (A9647; Sigma-Aldrich), 2.5 mM ATP (A2383; Sigma-Aldrich), 0.05 mM fructose 1,6-

bisphosphate (F6803; Sigma-Aldrich), 22.5 U myokinase from rabbit muscle (M3003; Sigma-Aldrich), 48 U pyruvate kinase from rabbit muscle (P1506; Sigma-Aldrich), and 24 U L-lactate dehydrogenase from rabbit muscle (10127230001; Roche) (or 45 U myokinase, 96 U pyruvate kinase, and 48 U L-lactate dehydrogenase for experiments in Fig. 4 B) in a buffer containing 150 mM NaCl and 25 mM Tris pH 7.4. The reaction was induced by the addition of 0.5 mM CoA (C3144; Sigma-Aldrich) or 0.5 mM AMP (01930; Sigma-Aldrich) for the positive control. The absorbance at 340 nm was measured for 15 min at 30°C with a Tecan SPARK Multimode Microplate Reader (RRID: SCR_021897; Tecan Life Sciences). The reaction for FadD activity was performed with 5 µg FadD and without liposomes. For the experiment in Fig. 4 B, Faa1-mCherry was used; for Fig. 4 C and Fig. S3 A, the EGFP-tagged variants were used; and for Fig. 5 B, the 10xHis-tagged versions were used. For normalization, the measurements were divided by the first timepoint before the initiation of the reaction, and the negative control was subtracted. For Fig. 5 B, the slope was calculated by creating a linear regression line with GraphPad Prism (RRID:SCR_002798) between 5 and 15 min after initiation.

### FRET-based assay

The incorporation of acyl-CoA was measured with a FRET-based assay. The reaction was mixed in a 50 µl volume containing 0.1 µM Faa1-WT-mCherry or Faa1-4D-mCherry, 60 µM Atg9-vesicle like liposomes containing lissamine rhodamine-DHPE (L-1392; Invitrogen) and NBD-DPPE (810144C; Avanti Polar Lipids, Inc.), 2.7 mM oleic acid (O1008; Sigma-Aldrich), 1 mM ATP (A2383; Sigma-Aldrich), 0.1 BSA (A9647; Sigma-Aldrich), and 10 mM $MgCl_2$ in buffer containing 200 mM NaCl and 25 mM Hepes pH 7.5. The reaction was started with the addition of 0.05 mM CoA (C3144; Sigma-Aldrich) or the same volume of buffer for the "–CoA" samples. Fluorescence intensity with an excitation wavelength of 485 nm and an emission wavelength of 535 nm was measured with a Tecan SPARK Multimode Microplate Reader (RRID:SCR_021897; Tecan Life Sciences) before the initiation of the reaction and 30 min after initiation while incubating at 30°C. For normalization, the measurements were divided by the measurement before initiation. Further, the fluorescence intensity of inactive Faa1 ("–CoA") was subtracted from the one of active Faa1 ("+CoA") to receive the increase in fluorescence intensity.

### Yeast strains and media

Strains for in vivo analyses of Faa1 localization and function derive from genetic modification of S. cerevisiae w303. Knockouts and tags were obtained via homologous recombination of PCR products of interest. Briefly, log-phase cells were washed with lithium acetate (100 mM) and incubated in a mix containing lithium acetate (L6883; Sigma-Aldrich), PEG 2000 (202509; Sigma-Aldrich), salmon sperm DNA (15632011; Invitrogen), and the PCR product of interest at 30°C for 30 min and then shifted to 42°C for 20 min.

Strains were grown in a synthetic complete medium (0.17% [wt/vol] yeast nitrogen base without amino acids and ammonium sulfate [233520; BD] supplemented with 0.5% [wt/vol]

ammonium sulfate [31119-M; Sigma-Aldrich], 2% [wt/vol] a-D-glucose monohydrate [16301; Sigma-Aldrich], and complete supplement [CSM] drop out [DCS0019; Formedium]). Log-phase cells were harvested, washed five times, and resuspended in SD-N medium (0.17% [wt/vol] yeast nitrogen base without amino acids and ammonium sulfate [233520; BD] and 2% [wt/vol] a-D-glucose monohydrate [16301; Sigma-Aldrich]). Based on the experiment, the SD-N medium was supplemented with cerulenin (219557, 50 µg/ml; Sigma-Aldrich) or glucose was reduced to 0.01% (wt/vol). Rapamycin (553210, 400 ng/ml; Calbiochem) was added to the synthetic complete medium.

### Cell viability
Cell suspension was mixed with 0.1% (vol/vol) trypan blue (T10282; Invitrogen) and incubated for 5 min. Cell viability was assessed by brightfield microscopy and expressed as a percentage of blue-stained (dead) in 100 cells.

### Fluorescent microscopy
Cells were imaged at room temperature in a 96-well plate with a glass bottom (655892; Greiner Bio-One) using an inverted microscope Zeiss Axio Observer Z1 equipped with an EC Plan-Neofluar 100×/1.3 Oil M27 objective, an EC Plan-Neofluar 40×/1.3 Oil RMS, an Orca Flash 4.0 LT+ camera, and a ZEN blue 3.3 pro software on Windows 11 pro 64-bit. Alternatively, experiments were performed with an inverted Nikon Ti2-E microscope (RRID:SCR_021068) equipped with a Yokogawa CSU-X1-A1 Nipkow spinning disk, a CFI Plan Apo λ 100×/1.45 Oil, WD 0.13 mm objective, a sCMOS back-illuminated Teledyne Prime BSI camera, and a VisiView 6.0 (Visitron Systems) software running on Windows 10 (64 bit). Images were deconvolved with Huygens Professional 23.04 (Scientific Volume Imaging) and analyzed using Fiji Version 2.9.0/1.53t (RRID:SCR_002285).

### Western blot and autophagic flux analysis
0.5 $OD_{600}$ yeast cells were collected and lysed with 0.255 M NaOH (S492060; Thermo Fisher Scientific). Proteins were precipitated with 100% trichloroacetic acid (T0699; Sigma-Aldrich) and washed once with cold acetone (CP40.3; Roth). Protein pellets were resuspended in protein loading buffer (62.5 mM Tris-HCl pH 6.8 [1018; GERBU], 10% glycerol [G5516; Sigma-Aldrich], 300 mM b-mercaptoethanol [8.05740; Sigma-Aldrich], 2% SDS [151213; GERBU], 0.1% bromphenol-Blue sod salt) and analyzed by SDS-PAGE using mouse anti-GFP antibody (Max Perutz Labs, Monoclonal antibody facility) in 3% (wt/vol) milk powder (T145; Roth) TBST-T (P7949; Sigma-Aldrich). Secondary antibody incubation was performed using Dylight 800 α-mouse (610145002; Rockland Immunochemicals). Detection and analysis were performed using the Li-COR Odyssey CLx fluorescence imager (RRID:SCR_014579; LI-COR Biosciences) and the Image Studio (RRID:SCR_015795, Version 2.1) software. Autophagic flux was assessed as the ratio between the free GFP and the total GFP signal (Free GFP and GFP Atg8 combined). Protein expression levels were normalized using Pgk1 as loading control (459250; Invitrogen). In Fig. S3 F, protein signal was normalized on a number of GFPs and Pgk1 level.

### 3D reconstruction of Faa1 on autophagosomes
3D reconstruction of Faa1 on autophagosomes was carried out using images of SMY511 cells after 1 h of Rapamycin treatment Rapamycin (553210, 400 ng/ml; Calbiochem). Autophagosomes were identified as Atg8 positive ring-like structures and saved as an independent picture. 3D reconstruction was performed using Mitograph in Terminal (-xy 0.056 -z 0.2), and mitopgraph files were visualized using ParaView (RRID:SCR_002516, v4.0).

### CHARM-GUI HMMM Faa1-membrane model
The Faa1-membrane interaction model was generated with CHARMM-GUI (RRID:SCR_025037) HMMM builder using the following membrane composition: 44% POPC, 41% POPI, 6% POPS, 6% POPE, and 3% PI3P. Faa1 structural information was gathered from Alphafold2. The default ion concentration was 0.15 M.

### Statistical analysis
All statistical analyses were performed with GraphPad Prism (RRID:SCR_002798). Data distribution was assumed to be normal but was not formally tested. The specific statistical tests used are stated in the figure legends. The ANOVA was performed as Holm-Šídák multiple comparison tests. The P values are indicated as $* < 0.05$, $** < 0.01$, $*** < 0.001$, $**** < 0.0001$.

### Online supplemental material
Fig. S1 shows supporting data for Fig. 2. Fig. S2 depicts the Alphafold model of human FACL4, the chromatograms of Faa1-WT-EGFP, Faa1-4D-EGFP, and Faa1-6D-EGFP, and the 3D construction of autophagosomes. Fig. S3 shows in vivo data supporting Fig. 4, a pelleting assay with 10x-His-tagged Faa1 variants, and a Western blot showing protein levels of genomically tagged Faa1-variants with and without the FYVE-domain. Table S1 lists the yeast strains used in this study. Table S2 lists the plasmids used in this study. Table S3 lists the lipids used for this study. Table S4 lists the liposomes prepared for this study.

### Data availability
Data reported in this study are available in the article and the supplementary material. Further information is available from the corresponding author.

## Acknowledgments
We thank Martin Graef for critical discussions and support.

This research was funded in whole or in part by the Austrian Science Fund (P35061-B, P32814-B). We thank the Max Perutz Labs BioOptics and Mass Spectrometry facility for technical support. Proteomics analyses were performed using the VBCF instrument pool. Open Access funding provided by the University of Vienna.

Author contributions: S. Martens conceptualized and supervised research; V. Baumann, S. Achleitner, and S. Tulli conceptualized research; and V. Baumann, S. Achleitner, S. Tulli, M. Schuschnig, and L. Klune performed research. All authors analyzed data and commented on the manuscript. S. Martens and V. Baumann wrote the manuscript.

Disclosures: All authors have completed and submitted the ICMJE Form for Disclosure of Potential Conflicts of Interest. S. Martens reported "S. Martens is member of the Scientific Advisory Board of Casma Therapeutics." No other disclosures were reported.

Submitted: 9 September 2023

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

# Supplemental material

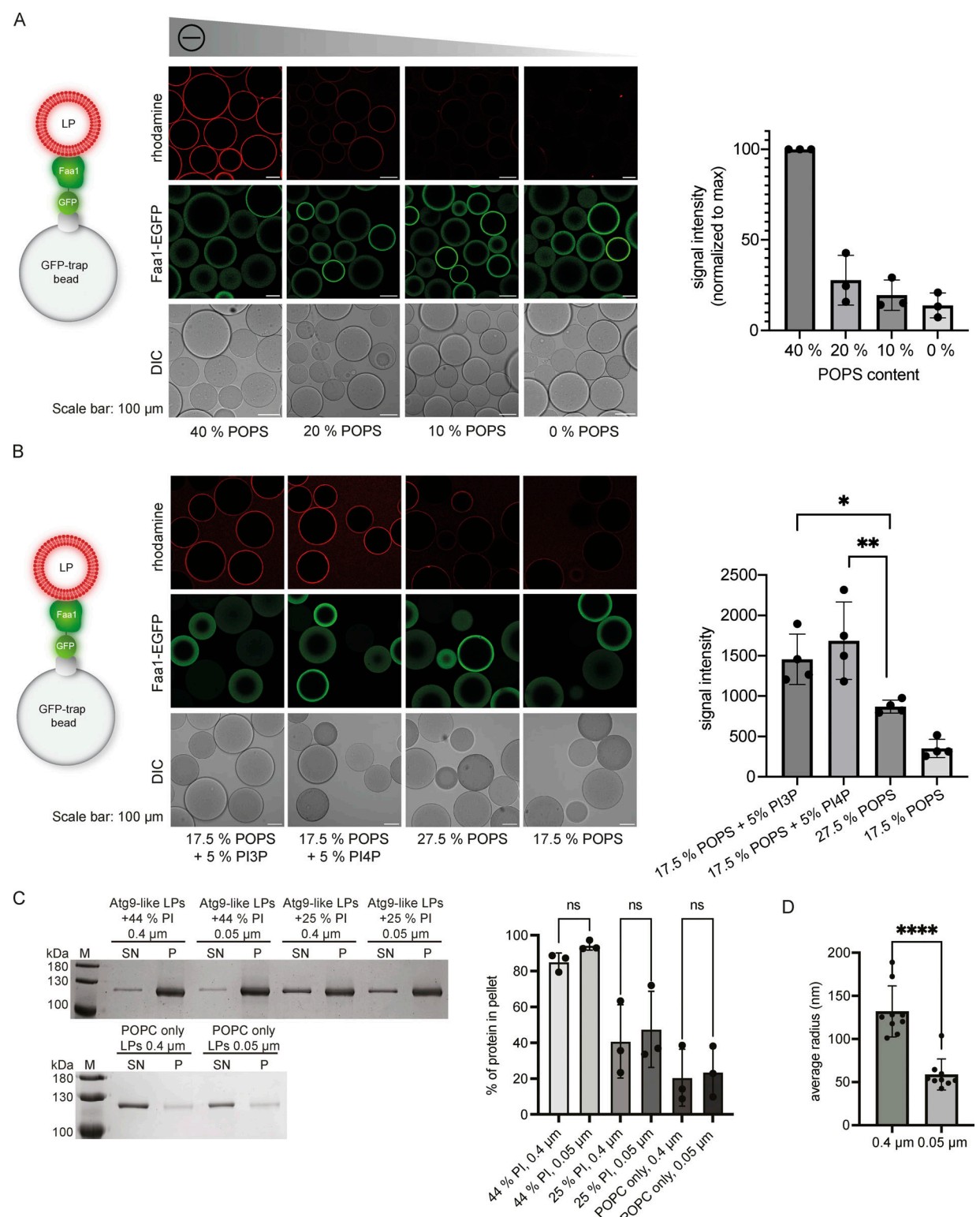

Figure S1.  **Interaction of Faa1 with membranes. (A and B)** Microscopy-based membrane recruitment assays analyzing Faa1 binding to different liposome compositions. Liposomes contained POPC and additionally, the amount of POPS, PI3P, or PI4P indicated in the figures and 0.5% lissamine rhodamine-DHPE. Quantifications show mean ± SD ($n$ = 3). LP = liposome. Two-sided $t$ test: * < 0.05, ** < 0.01. **(C)** Cosedimentation assay directly comparing Faa1 binding to differently sized liposomes to test for curvature dependency including quantification. Liposome composition resembles Atg9 vesicles with 44% POPC, 6% POPS, 44% liver PI, 6% POPE, and 0.5% lissamine rhodamine-DHPE, whereas liver PI was substituted with POPC in "Atg9-like LPs + 25% PI" and for "POPC only" liposomes contained 99.5% POPC and 0.5% lissamine rhodamine-DHPE. Data are means ± SD ($n$ = 3). One-way ANOVA: ns = not significant. LP = liposome, SN = supernatant, P = pellet. **(D)** Dynamic light scattering evaluation of the average radius of liposomes from C. When labeled "0.4 µm" the liposomes were extruded only through a 0.4 µm filter while those labeled "0.05 µm" were additionally extruded through a 0.05 µm filter. Two-sided $t$ test: **** < 0.0001. Source data are available for this figure: SourceData FS1.

Figure S2. **Conservation, purification, localization, and expression of Faa1. (A)** Alphafold model of human FACL4. The distribution of the electrostatic potentials was calculated with the APBS method on the molecular surface of FACL4. Electropositively and electronegatively charged areas are colored in blue and red, respectively. Neutral residues are depicted in white. **(B)** The size exclusion chromatograms of Faa1-WT-EGFP, Faa1-4D-EGFP, and Faa1-6D-EGFP show that all three proteins elute from the column at the same volume. **(C)** 3D reconstruction of autophagosomes from different angles. Atg8 (magenta) and Faa1 (cyan). Examples obtained from cells from Fig. 3 E. **(D)** Normalized protein levels of genomically tagged Faa1-WT, Faa1-4D, and Faa1-6D by Western blot quantification. Data are means ± SD ($n$ = 3). Source data are available for this figure: SourceData FS2.

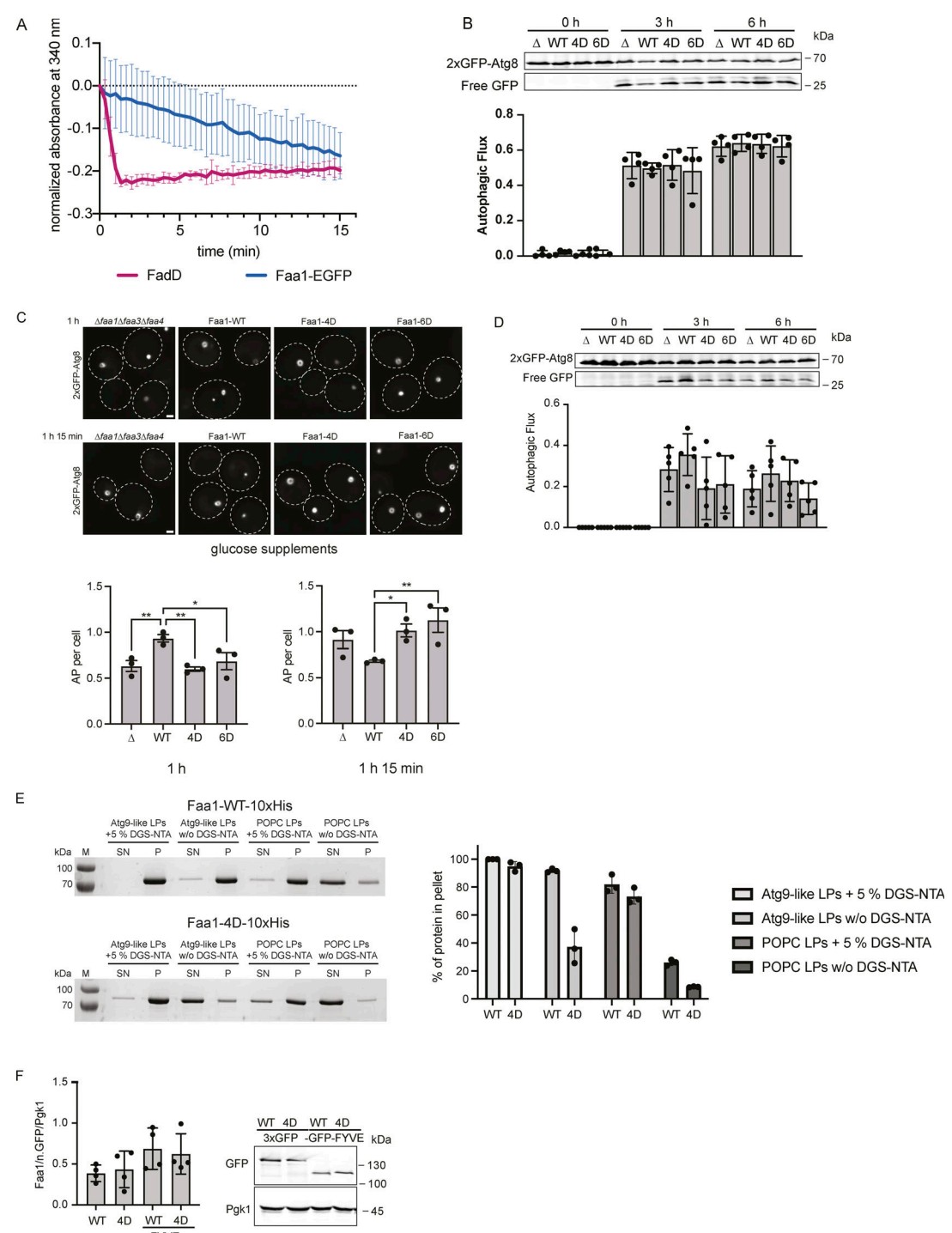

Figure S3.  **Activity, membrane recruitment, and expression of Faa1. (A)** Coupled enzymatic assay comparing the activities of FadD in the absence of liposomes and Faa1 in the presence of Atg9 vesicle-like liposomes (44% POPC, 6% POPS, 41.5% liver PI, 6% POPE, 2.5% PI3P). Data are means ± SD ($n$ = 3). **(B)** The autophagic flux of indicated strains expressing 2xGFP-Atg8 after 0, 3, and 6 h of starvation. Upon fusion of the autophagosome with the vacuole to form an autolysosome, Atg8 is rapidly degraded while GFP remains intact for a longer time and can be detected as free GFP by Western blot analysis. Data are means ± SD ($n$ = 4). **(C)** Fluorescent microscopy assessment of autophagosome biogenesis in indicated strains after 1 and 1 h 15 min of nitrogen starvation and glucose depletion (0.01% wt/vol). Top, representative images of the analyzed strains at indicated time points. Dashed lines indicate cell boundaries. Scale bar, 1 μm. Bottom, respective quantification of AP. Data are means ± SEM ($n$ = 3; 50 cells/strain per replicate). One-way ANOVA: * < 0.05, ** < 0.01. AP = autophagosome. **(D)** Autophagic flux of indicated strains expressing 2xGFP-Atg8 after 0, 3, and 6 h of starvation and glucose depletion (0.01% wt/vol). Data are means ± SD ($n$ = 5). **(E)** Cosedimentation assay of Faa-WT-10xHis and Faa1-4D-10xHis with Atg9 vesicle-like LPs or POPC LPs with or without 5% DGS-NTA lipids including quantification. Liposome composition resembled Atg9 vesicles ("Atg9-like LPs") with 44% POPC, 6% POPS, 41.5% liver PI, 6% POPE, and 2.5% PI3P or only contained 100% POPC ("POPC LPs"). For liposomes containing 5% DGS-NTA, the percentage of POPC was reduced. Data are means ± SD ($n$ = 3). LP = liposome, SN = supernatant, P = pellet. **(F)** Normalized protein levels of genomically tagged Faa1-WT, Faa1-4D, Faa1-WT-FYVE, and Faa1-4D-FYVE by Western blot quantification. Data are means ± SD ($n$ = 4). Source data are available for this figure: SourceData FS3.

Provided online are Table S1, Table S2, Table S3, and Table S4. Table S1 shows yeast strains used in this study. Table S2 shows plasmids used in this study. Table S3 shows lipids used in this study. Table S4 shows liposomes prepared for this study.

