## [Peer Review File · The Journal of Cell Biology]

Faa1 membrane binding drives positive feedback in autophagosome biogenesis via fatty acid activation

Verena Baumann, Sonja Achleitner, Susanna Tulli, Martina Schuschnig, Lara Klune, and Sascha Martens

Corresponding Author(s): Sascha Martens, University of Vienna, Max Perutz Labs

Review Timeline:

Submission Date:	2023-09-09
Editorial Decision:	2023-10-11
Revision Received:	2024-02-14
Editorial Decision:	2024-03-11
Revision Received:	2024-03-21

Monitoring Editor: William Prinz

Scientific Editor: Andrea Marat

Transaction Report:

DOI: <https://doi.org/10.1083/jcb.202309057>

October 11, 2023

Re: JCB manuscript #202309057

Prof. Sascha Martens
University of Vienna, Max Perutz Labs
Dr Bohr-Gasse 9/3
Vienna 1030
Austria

Dear Prof. Martens,

Thank you for submitting your manuscript entitled "Faa1 membrane binding drives positive feedback in autophagosome biogenesis via fatty acid activation". The manuscript has been evaluated by expert reviewers, whose reports are appended below. Unfortunately, after an assessment of the reviewer feedback, our editorial decision is against publication in JCB.

You will see that while all three reviewers found your study well done and largely convincing, they all share the opinion that it is not yet a significant enough advance to be appropriate for JCB. We agree. The reviewers ask for more mechanistic insight into how Faa1 is targeted to autophagosomes or how membrane binding impacts enzymatic activity. They have some constructive suggestions for how to do this. We would welcome a resubmission that addresses either of these concerns.

Although your manuscript is intriguing, I feel that the points raised by the reviewers are more substantial than can be addressed in a typical revision period. If you wish to expedite publication of the current data, it may be best to pursue publication at another journal.

Given interest in the topic, I would be open to resubmission to JCB of a significantly revised and extended manuscript that fully addresses the reviewers' concerns and is subject to further peer-review. If you would like to resubmit this work to JCB, please contact the journal office to discuss an appeal of this decision or you may submit an appeal directly through our manuscript submission system. Please note that priority and novelty would be reassessed at resubmission.

Regardless of how you choose to proceed, we hope that the comments below will prove constructive as your work progresses. We would be happy to discuss the reviewer comments further once you've had a chance to consider the points raised in this letter. You can contact the journal office with any questions at cellbio@rockefeller.edu.

Thank you for thinking of JCB as an appropriate place to publish your work.

Sincerely,

William Prinz, PhD
Monitoring Editor

Andrea L. Marat, PhD
Senior Scientific Editor

Journal of Cell Biology

Reviewer #1 (Comments to the Authors (Required)):

This manuscript is a straightforward characterization of an electrostatic protein-membrane interaction. The authors set out to understand how Faa1 is recruited to developing autophagosomes. Although previous studies in cells had indicated a key role for several autophagy proteins in this recruitment, the authors discover that membrane interaction *in vitro* is entirely dependent upon the presence of sufficient negative charge in the membrane. They test membranes of varying complexity, including isolated yeast Atg9 vesicles (seeds of the autophagosome), proteoliposomes reconstituted with Atg9, and pure lipid liposomes. There is no evidence that the presence or absence of Atg9 (or any other protein) is important to the activity they describe. Perhaps most interesting, they explore the activity of Faa1 and discover that it depends upon membrane binding. Mutations which disrupt the electrostatics of Faa1 also disrupt its membrane recruitment and its ability to support autophagy in cells. Note, these mutations block Faa1 recruitment to all membranes in cells, including the plasma membrane, indicating a key role for these amino acids in protein function or structure but not autophagic specificity.

The experiments are well conceived, and the conclusion of an electrostatic interaction is convincing. The scope of the paper is somewhat narrow however, further understanding of how Faa1 is specifically targeted to autophagosomes and/or the role of other known proteins including Atg9, Atg14, in that targeting would significantly elevate the impact of the story.

Reviewer #2 (Comments to the Authors (Required)):

A few years ago, the acyl-CoA synthetase Faa1 was found to be recruited to phagophore membrane to allow efficient membrane expansion by providing substrate for lipid neosynthesis in the ER in yeast. However, the mechanisms involved in the regulation of Faa1 targeting and activity are still unknown. In their work, Baumann and colleagues showed in vitro that Faa1 has the ability to bind directly to liposomes enriched in negatively charged lipids without the requirement of any protein partner. They identified Faa1 positively charged surface and mutated Lys residues to show that these regions are involved in the binding to negatively charged membranes. Then they show that membrane binding is important for Faa1 activity in contrast to the bacterial counterpart FadD. Finally, they confirm in vivo that the positively charged surface of Faa1 is required for phagophore binding, expansion and cell survival. Thus, the results of this study provide an important step toward the understanding of the key steps required for phagophore expansion by dissecting the mechanisms involved in Faa1 membrane binding and activity regulation. The experiments are well designed and the conclusions are well supported by the results. Whereas the ability of Faa1 to bind negatively charged membranes via positively charged surface both in vivo and in vitro is highly convincing, I would like to suggest to the authors to add a couple of experiments to further investigate what are the factors essential for Faa1 activity regulation (related to experiments in Fig. 3), which is an important point to my opinion.

Major

Related to experiments performed in Fig. 3, it is not clear to me what are the parameters required for the activity. Is it just the binding to membrane and/or the negative charges on the membrane and/or the Lys residues of Faa1 that have been mutated? To demonstrate if only lipid binding or also the presence of negative charges is required, I will suggest to perform the activity test with a His-tagged version of the WT Faa1 and liposomes containing 95%POPC + 5% DGS-NiNTA to artificially tether the protein to liposomes. If Faa1 is functional in this setup, this means that only binding is required for activity, if no activity is detected, it will suggest that both the binding and the negatively charged lipids are important to regulate Faa1 activity. Concerning the role of the Faa1 Lys residues, the fact that the 4D mutant still show binding to liposomes (reduced by 50% in both liposomes co-sedimentation assays) but an activity close to negative control suggests that those Lys are important not only for binding but also for activity. To clearly answer this question, I will also suggest to express the 4D or 6D mutant with a His-tag to artificially tether the protein to ATG9-like liposomes with DGS-NiNTA and see whether in this condition the activity is restored or not. This kind of experiments can be helpful to go closer into the mechanisms involved in Faa1 activity regulation and membrane expansion.

Minor

- Can the authors make a quantification of the images they presented in Fig 4A in order to clearly show what is the proportion of ATG8 vesicles labelled with Faa1 WT, 4D and 6D?
- On Fig. 4C and E, can the authors indicate with arrows the ATG8 structures they have considered as autophagosomes for the quantification?
- Can the authors add ATG9 on the scheme Fig. 1A?

Reviewer #3 (Comments to the Authors (Required)):

This study dissects the role of fatty acyl CoA ligase FAA1p in autophagosome biogenesis. Using AlphaFold predictions, it finds FAA1p directly binds membranes via a charged region, and this membrane recruitment is necessary for it to function as an enzyme producing FA-CoAs for membrane biosynthesis. Using a liposome tethering system, they show that a positively charged region on FAA1p interacts with negatively charged lipids on liposomes. Mutation of this region impacts FAA1p recruitment to membranes as well as recruitment to organelles in yeast. These mutants also show defects in enzymatic activity, indicating that FAA1p requires membrane recruitment for function.

This is a straightforward and generally well organized study that nicely combines in vitro and in vivo experiments to functionally dissect how membrane targeting of FAA1p influences its role in autophagosome maturation. In general the experiments are well designed and the data appear rigorously obtained. Further mechanistic dissection of exactly how membrane recruitment of FAA1p impacts its activity and function would strengthen this study.

Specific comments:

- a) The study nicely utilizes membrane binding assays combined with enzymatic activity assays to investigate how FAA1p

localizes to membranes, and how this impacts enzymatic activity. An open question is whether FAA1p requires membrane localization in order to better access fatty acid substrates, or to more easily release fatty acyl-CoA enzymatic products. Fatty acyl-CoAs have been proposed to insert into membrane bilayers, which would make membrane recruitment more favorable for FAA1p activity. Delineating these possibilities would mechanistically enhance the study. One possibility is to bind FAA1p to beads alone and see if this also enhances enzymatic activity independent of membrane lipids.

b)Based on the Alphafold structure, it is expected that FAA1p would not show any membrane curvature preference. However, curvature preference for membrane binding might be an excellent way of regulating FAA1p's impact on phagophore expansion. It would be interesting to check for curvature binding preference by conducting liposome binding assays with liposomes of different sizes/curvatures.

c)If the model is correct, the 4D or 6D mutants could be rescued in vivo by adding an electrostatically charged or other membrane binding moiety to Faa1.

We thank the reviewers for the positive and constructive comments on our manuscript. As evident from the point-by-point reply, we have taken them very seriously and addressed all major comments experimentally.

Reviewer 1

This manuscript is a straightforward characterization of an electrostatic protein-membrane interaction. The authors set out to understand how Faa1 is recruited to developing autophagosomes. Although previous studies in cells had indicated a key role for several autophagy proteins in this recruitment, the authors discover that membrane interaction *in vitro* is entirely dependent upon the presence of sufficient negative charge in the membrane. They test membranes of varying complexity, including isolated yeast Atg9 vesicles (seeds of the autophagosome), proteoliposomes reconstituted with Atg9, and pure lipid liposomes. There is no evidence that the presence or absence of Atg9 (or any other protein) is important to the activity they describe. Perhaps most interesting, they explore the activity of Faa1 and discover that it depends upon membrane binding. Mutations which disrupt the electrostatics of Faa1 also disrupt its membrane recruitment and its ability to support autophagy in cells. Note, these mutations block Faa1 recruitment to all membranes in cells, including the plasma membrane, indicating a key role for these amino acids in protein function or structure but not autophagic specificity.

The experiments are well conceived, and the conclusion of an electrostatic interaction is convincing. The scope of the paper is somewhat narrow however, further understanding of how Faa1 is specifically targeted to autophagosomes and/or the role of other known proteins including Atg9, Atg14, in that targeting would significantly elevate the impact of the story.

We thank the reviewer for the constructive criticism. We agree that identifying the specific mechanisms underlying Faa1 localization to autophagosomes is highly interesting. Therefore, we conducted further experiments to identify potential recruitment factors. As shown in the manuscript and also stated by the reviewer, Atg9 is not sufficient to recruit Faa1 to liposomes (Figure 1C). Additionally, we tested numerous core autophagy factors (Atg1-Atg13, Atg2-18, Atg21, Atg12-Atg5-Atg16, Atg3, Atg7, Atg11, Atg8) for a possible direct interaction with Faa1. However, none of the used autophagy proteins bound to Faa1 under the tested conditions (Rebuttal Figure 1, below).

Further liposome co-sedimentation assays yielded deeper insights into Faa1's membrane binding preferences. Figure 2C shows that under more stringent conditions, Faa1 binding to liposomes containing phosphoinositides like PI3P and PI4P is strongly enhanced compared to liposomes with the same net charge but without PI3P and PI4P. This was confirmed by testing Faa1 binding to giant unilamellar vesicles (GUVs) (Figure 2D). While Faa1 robustly localized to PI3P positive GUVs, it could not be detected on GUVs with the same net charge but lacking PI3P. Based on these findings we hypothesize that Faa1 is recruited to nucleated phagophores mainly by their particularly high negative charge. The observation that Faa1 bound to small negatively charged liposomes without PI3P but not to GUVs of the same composition further suggests a contribution of membrane curvature to Faa1 recruitment. As autophagy proceeds, Atg2 and Vps13 transfer lipids from the ER to the phagophore, leading to an expansion of the membrane and a dilution of the highly concentrated negative charges (Dabrowski et al., 2023; Sawa-Makarska et al., 2020). To ensure continuous Faa1 recruitment to the growing phagophore, the Atg14 containing PI3KC1C3 is required to generate PI3P.

Rebuttal Figure 1. Coomassie stained SDS-PAGE gels showing pull-downs with Faa1-mCherry on RFP-trap beads as a bait and several different proteins of the core autophagy machinery as prey. As a negative control RFP-trap beads without Faa1 were used. In: input, b: bead-bound. Arrows indicate respective protein.

Reviewer 2

A few years ago, the acyl-CoA synthetase Faa1 was found to be recruited to phagophore membrane to allow efficient membrane expansion by providing substrate for lipid neosynthesis in the ER in yeast. However, the mechanisms involved in the regulation of Faa1 targeting and activity are still unknown. In their work, Baumann and colleagues showed in vitro that Faa1 has the ability to bind directly to liposomes enriched in negatively charged lipids without the requirement of any protein partner. They identified Faa1 positively charged surface and mutated Lys residues to show that these regions are involved in the binding to negatively charged membranes. Then they show that membrane binding is important for Faa1 activity in contrast to the bacterial counterpart FadD. Finally, they confirm in vivo that the positively charged surface of Faa1 is required for phagophore binding, expansion and cell survival. Thus, the results of this study provide an important step toward the understanding of the key steps required for phagophore expansion by dissecting the mechanisms involved in Faa1 membrane binding and activity regulation. The experiments are well designed and the conclusions are well supported by the results. Whereas the ability of Faa1 to bind negatively charged membranes via positively charged surface both in vivo and in vitro is highly convincing, I would like to suggest to the authors to add a couple of experiments to further investigate what are the factors essential for Faa1 activity regulation (related to experiments in Fig. 3), which is an important point to my opinion.

Related to experiments performed in Fig. 3, it is not clear to me what are the parameters required for the activity. Is it just the binding to membrane and/or the negative charges on the membrane and/or the Lys residues of Faa1 that have been mutated? To demonstrate if only lipid binding or also the presence of negative charges is required, I will suggest to perform the activity test with a His-tagged version of the WT Faa1 and liposomes containing 95%POPC + 5% DGS-NiNTA to artificially tether the protein to liposomes. If Faa1 is functional in this setup, this means that only binding is required for activity, if no activity is detected, it will suggest that both the binding and the negatively charged lipids are important to regulate Faa1 activity. Concerning the role of the Faa1 Lys residues, the fact that the 4D mutant still show binding to liposomes (reduced by 50% in both liposomes co-sedimentation assays) but an activity close to negative control suggests that those Lys are important not only for binding but also for activity. To clearly answer this question, I will also suggest to express the 4D or 6D mutant with a His-tag to artificially tether the protein to ATG9-like liposomes with DGS-NiNTA and see whether in this condition the activity is restored or not. This kind of experiments can be helpful to go closer into the mechanisms involved in Faa1 activity regulation and membrane expansion.

We thank the reviewer for the positive feedback and the points raised. Following the constructive suggestions, we carried out enzymatic assays using 10xHis-tagged Faa1 in combination with DGS-NTA lipids (Figure 5B, S5E). This set up allowed us to test whether spatial proximity of Faa1 to membranes would be sufficient to enhance its enzymatic activity or if direct interaction of the identified surface area is indeed required for efficient Faa1 function. Figure 5B shows a comparison between Faa1-Wt and 4D with and without artificial tether. For easier comparison between the numerous different set ups, we decided to display the slope of the enzymatic curves instead of the curves themselves. Interestingly, bringing Faa1 to the membrane artificially did not enhance its enzymatic activity compared to the non-tethered version, underlining the relevance of the direct interaction of the protein with the

membrane. In addition, we have tested if artificial recruitment of Faa1 to phagophores in cells can restore activity of the Faa1 4D mutant and found that this also did not rescue its activity (Figure 5C-E, S5F).

Can the authors make a quantification of the images they presented in Fig 4A in order to clearly show what is the proportion of ATG8 vesicles labelled with Faa1 WT, 4D and 6D?

On Fig. 4C and E, can the authors indicate with arrows the ATG8 structures they have considered as autophagosomes for the quantification?

Can the authors add ATG9 on the scheme Fig. 1A?

The reviewer's suggestions are very much appreciated, and we addressed all minor comments above in the revised manuscript.

Reviewer 3

This study dissects the role of fatty acyl CoA ligase FAA1p in autophagosome biogenesis. Using AlphaFold predictions, it finds FAA1p directly binds membranes via a charged region, and this membrane recruitment is necessary for it to function as an enzyme producing FA-CoAs for membrane biosynthesis. Using a liposome tethering system, they show that a positively charged region on FAA1p interacts with negatively charged lipids on liposomes. Mutation of this region impacts FAA1p recruitment to membranes as well as recruitment to organelles in yeast. These mutants also show defects in enzymatic activity, indicating that FAA1p requires membrane recruitment for function.

This is a straightforward and generally well organized study that nicely combines in vitro and in vivo experiments to functionally dissect how membrane targeting of FAA1p influences its role in autophagosome maturation. In general the experiments are well designed and the data appear rigorously obtained. Further mechanistic dissection of exactly how membrane recruitment of FAA1p impacts its activity and function would strengthen this study.

We are grateful for the reviewers' thorough assessment of our manuscript and for the positive feedback.

The study nicely utilizes membrane binding assays combined with enzymatic activity assays to investigate how FAA1p localizes to membranes, and how this impacts enzymatic activity. An open question is whether FAA1p requires membrane localization in order to better access fatty acid substrates, or to more easily release fatty acyl-CoA enzymatic products. Fatty acyl-CoAs have been proposed to insert into membrane bilayers, which would make membrane recruitment more favorable for FAA1p activity. Delineating these possibilities would mechanistically enhance the study. One possibility is to bind FAA1p to beads alone and see if this also enhances enzymatic activity independent of membrane lipids.

To address the reviewers' question how Faa1 membrane binding facilitates its enzymatic activity we conducted FRET assays comparable to lipid transfer/fusion assays. By adding Faa1 +/- CoA to NBD-DPPE and lissamine rhodamine-DHPE labelled liposomes we tested for direct membrane incorporation of the produced acyl-CoA. We chose this approach over the suggested bead experiment since we think that this is a more direct way to decipher the role of membrane binding for the protein's activity. Figure 5A shows the increased NBD-DPPE signal before and after addition of CoA caused by the raising distance between the fluorophores. This result suggests that Faa1 localization on membranes facilitates an efficient release of its product into the membrane and thereby provides a deeper insight into the mechanism underlying Faa1 activity.

Based on the AlphaFold structure, it is expected that FAA1p would not show any membrane curvature preference. However, curvature preference for membrane binding might be an excellent way of regulating FAA1p's impact on phagophore expansion. It would be interesting to check for curvature binding preference by conducting liposome binding assays with liposomes of different sizes/curvatures.

We thank the reviewer for this comment and agree that membrane curvature could represent another important regulation step for Faa1 function. Accordingly, we conducted membrane-protein interaction assays with membranes of different sizes. When comparing Faa1's recruitment to small unilamellar vesicles we could not observe any preference of Faa1 towards higher curved membranes (Figure S1C-D). A 3D reconstruction of microscopy images observing Faa1 shows the protein being distributed all over the autophagosome (Figure S2C). We proceeded to conduct experiments with GUVs, which are essentially flat. These experiments suggest that also membrane curvature might have a stimulatory effect on Faa1 recruitment because in this experimental condition the incorporation of PI3P was essential for membrane the recruitment of Faa1 (Figure 2D). Taken together, this leads us to the hypothesis, that after initial Faa1 recruitment to Atg9 vesicles and very early phagophores, mediated by a high PI content and high curvature, continuous Faa1 targeting to the expanding phagophore relies on an active PI3KC1C3 producing PI3P. The synthesis of PI3P in turn overrides the preference of Faa1 for curved membranes.

If the model is correct, the 4D or 6D mutants could be rescued *in vivo* by adding an electrostatically charged or other membrane binding moiety to Faa1.

Following the reviewer's comment, we tested whether the phenotypes associated with the Faa1-4D and 6D mutants could be rescued *in vivo* by relocating Faa1 back to membranes. To this end, we generated constructs expressing Faa1 fused to a PI3P-specific FYVE domain. We observed localization of Faa1-4D to Atg8-positive structures and the vacuolar membrane (Figure 5C). Despite comparable protein expression levels (Figure S3F), the restoration of cell survival was only minimal (Figure 5C) and autophagy could not be rescued (Figure 5 D-E). This underlines that the correct orientation of the enzyme via the identified membrane interaction surface is key for its functionality in cells.

References

- Dabrowski, R., S. Tulli, and M. Graef. 2023. Parallel phospholipid transfer by Vps13 and Atg2 determines autophagosome biogenesis dynamics. *J Cell Biol.* 222.
- Sawa-Makarska, J., V. Baumann, N. Coudevylle, S. von Bulow, V. Nogellova, C. Abert, M. Schuschnig, M. Graef, G. Hummer, and S. Martens. 2020. Reconstitution of autophagosome nucleation defines Atg9 vesicles as seeds for membrane formation. *Science.* 369.

March 11, 2024

RE: JCB Manuscript #202309057R-A

Prof. Sascha Martens
University of Vienna, Max Perutz Labs
Dr Bohr-Gasse 9/3
Vienna 1030
Austria

Dear Prof. Martens:

Thank you for submitting your revised manuscript entitled "Faa1 membrane binding drives positive feedback in autophagosome biogenesis via fatty acid activation". As you will see the reviewers now support publication, therefore we would be happy to publish your paper in JCB pending final revisions necessary to meet our formatting guidelines (see details below).

A. MANUSCRIPT ORGANIZATION AND FORMATTING:

- 1) Text limits: Character count for Reports is < 20,000, not including spaces. Count includes abstract, introduction, * combined results and discussion, and acknowledgments. Count does not include title page, figure legends, materials and methods, references, tables, or supplemental legends.
- 2) Figures limits: Reports may have up to 5 main text figures.
- 3) Figure formatting: Scale bars must be present on all microscopy images, including inset magnifications. Molecular weight or nucleic acid size markers must be included on all gel electrophoresis. In order to accommodate readers with red-green color blindness, we suggest that you avoid the used of red/green color schemes.
- 4) Statistical analysis: Error bars on graphic representations of numerical data must be clearly described in the figure legend. The number of independent data points (n) represented in a graph must be indicated in the legend. Statistical methods should be explained in full in the materials and methods. For figures presenting pooled data the statistical measure should be defined in the figure legends. Please also be sure to indicate the statistical tests used in each of your experiments (either in the figure legend itself or in a separate methods section) as well as the parameters of the test (for example, if you ran a t-test, please indicate if it was one- or two-sided, etc.). Also, if you used parametric tests, please indicate if the data distribution was tested for normality (and if so, how). If not, you must state something to the effect that "Data distribution was assumed to be normal but this was not formally tested."
- 5) Abstract and title: The abstract should be no longer than 160 words and should communicate the significance of the paper for a general audience. The title should be less than 100 characters including spaces. Make the title concise but accessible to a general readership.
- 6) Materials and methods: Should be comprehensive and not simply reference a previous publication for details on how an experiment was performed. Please provide full descriptions in the text for readers who may not have access to referenced manuscripts. * For example, please describe the method used to produce GUVs.*
- 7) *All antibodies, cell lines, animals, and tools used in the manuscript should be described in full, including accession numbers for materials available in a public repository such as the Resource Identification Portal. Please be sure to provide the sequences for all of your primers/oligos and RNAi constructs in the materials and methods. You must also indicate in the methods the source, species, and catalog numbers (where appropriate) for all of your antibodies. Please also indicate the acquisition and quantification methods for immunoblotting/western blots.*
- 8) Microscope image acquisition: The following information must be provided about the acquisition and processing of images:
 - a. Make and model of microscope
 - b. Type, magnification, and numerical aperture of the objective lenses
 - c. Temperature
 - d. Imaging medium

- e. Fluorochromes
- f. Camera make and model
- g. Acquisition software
- h. Any software used for image processing subsequent to data acquisition. Please include details and types of operations involved (e.g., type of deconvolution, 3D reconstitutions, surface or volume rendering, gamma adjustments, etc.).

10) Supplemental materials: There are strict limits on the allowable amount of supplemental data. Reports may have up to 3 supplemental figures. Please also note that tables, like figures, should be provided as individual, editable files. A summary of all supplemental material should appear at the end of the Materials and methods section.

13) ORCID IDs: ORCID IDs are unique identifiers allowing researchers to create a record of their various scholarly contributions in a single place. Please note that ORCID IDs are now *required* for all authors. At resubmission of your final files, please be sure to provide your ORCID ID and those of all co-authors.

Please note that JCB now requires authors to submit Source Data used to generate figures containing gels and Western blots with all revised manuscripts. This Source Data consists of fully uncropped and unprocessed images for each gel/blot displayed in the main and supplemental figures. Since your paper includes cropped gel and/or blot images, please be sure to provide one Source Data file for each figure that contains gels and/or blots along with your revised manuscript files. File names for Source Data figures should be alphanumeric without any spaces or special characters (i.e., SourceDataF#, where F# refers to the associated main figure number or SourceDataFS# for those associated with Supplementary figures). The lanes of the gels/blots should be labeled as they are in the associated figure, the place where cropping was applied should be marked (with a box), and molecular weight/size standards should be labeled wherever possible.

Journal of Cell Biology now requires a data availability statement for all research article submissions. These statements will be published in the article directly above the Acknowledgments. The statement should address all data underlying the research presented in the manuscript. Please visit the JCB instructions for authors for guidelines and examples of statements at (<https://rupress.org/jcb/pages/editorial-policies#data-availability-statement>).

B. FINAL FILES:

****It is JCB policy that if requested, original data images must be made available to the editors. Failure to provide original images upon request will result in unavoidable delays in publication. Please ensure that you have access to all original data images prior to final submission.****

****The license to publish form must be signed before your manuscript can be sent to production. A link to the electronic license to publish form will be sent to the corresponding author only. Please take a moment to check your funder requirements before choosing the appropriate license.****

Thank you for your attention to these final processing requirements. Please revise and format the manuscript and upload materials within 7 days. If you need an extension for whatever reason, please let us know and we can work with you to determine a suitable revision period.

Thank you for this interesting contribution, we look forward to publishing your paper in Journal of Cell Biology.

Sincerely,

William Prinz, PhD
Monitoring Editor

Andrea L. Marat, PhD
Senior Scientific Editor

Journal of Cell Biology

Reviewer #2 (Comments to the Authors (Required)):

The authors responded to all my concerns and the additional experiments they performed significantly improved the manuscript and the understanding of the factors that regulate the localization and activity of Faa1. It's a very nice paper.

Reviewer #3 (Comments to the Authors (Required)):

The revision addresses the majority of concerns raised. While the work remains somewhat limited in scope, it provides mechanistically important additions to our understanding of Faa1 and its role in autophagosome biogenesis.